EMBO
Molecular Medicine

# Brain-targeted stem cell gene therapy corrects mucopolysaccharidosis type II via multiple mechanisms

Hélène FE Gleitz [iD], Ai Yin Liao, James R Cook [iD], Samuel F Rowlston, Gabriella MA Forte, Zelpha D'Souza, Claire O'Leary, Rebecca J Holley & Brian W Bigger* [iD]

## Abstract

The pediatric lysosomal storage disorder mucopolysaccharidosis type II is caused by mutations in IDS, resulting in accumulation of heparan and dermatan sulfate, causing severe neurodegeneration, skeletal disease, and cardiorespiratory disease. Most patients manifest with cognitive symptoms, which cannot be treated with enzyme replacement therapy, as native IDS does not cross the blood–brain barrier. We tested a brain-targeted hematopoietic stem cell gene therapy approach using lentiviral IDS fused to ApoEII (IDS.ApoEII) compared to a lentivirus expressing normal IDS or a normal bone marrow transplant. In mucopolysaccharidosis II mice, all treatments corrected peripheral disease, but only IDS.ApoEII mediated complete normalization of brain pathology and behavior, providing significantly enhanced correction compared to IDS. A normal bone marrow transplant achieved no brain correction. Whilst corrected macrophages traffic to the brain, secreting IDS/IDS.ApoEII enzyme for cross-correction, IDS.ApoEII was additionally more active in plasma and was taken up and transcytosed across brain endothelia significantly better than IDS via both heparan sulfate/ApoE-dependent receptors and mannose-6-phosphate receptors. Brain-targeted hematopoietic stem cell gene therapy provides a promising therapy for MPS II patients.

**Keywords** apolipoprotein E; blood–brain barrier; Hunter; mucopolysaccharidosis type II; stem cell gene therapy

**Subject Categories** Genetics, Gene Therapy & Genetic Disease; Neuroscience; Stem Cells

## Introduction

Mucopolysaccharidosis type II (MPS II, OMIM #309900), or Hunter syndrome, is a pediatric X-linked lysosomal storage disorder caused by mutations in the *IDS* gene, leading to deficiencies in iduronate-2-sulfatase enzyme (EC 3.1.6.13). IDS insufficiency affects the catabolism of both heparan sulfate (HS) and dermatan sulfate (DS), leading to their accumulation in all cells (Neufeld & Muenzer, 2001). MPS II affects 1.3 per 100,000 male live births (Poorthuis *et al*, 1999; Baehner *et al*, 2005; Wraith *et al*, 2008) and is a chronic and progressive multisystem disease affecting a multitude of organs, which forms a continuum between attenuated and severe forms. Clinical manifestations in MPS II include skeletal abnormalities, known as dysostosis multiplex, short stature, joint stiffness, and hepatosplenomegaly, accompanied by cardiorespiratory symptoms (Cardone *et al*, 2006; Wraith *et al*, 2008). Severe MPS II, which is most common, also features progressive neurodegeneration, followed by death in teenage years due to obstructive airway disease and cardiac failure (Meikle *et al*, 1999; Wraith *et al*, 2008; Holt *et al*, 2011).

Enzyme replacement therapy (ERT), where exogenous replacement enzyme is delivered intravenously and internalized by cells via the mannose-6-phosphate (M6P) receptor, has been used to treat the somatic symptoms in MPS II patients (Muenzer *et al*, 2006; Eng *et al*, 2007). However, enzyme circulating in the bloodstream is prevented from reaching the central nervous system (CNS) by the blood–brain barrier (BBB), considerably reducing therapeutic benefits for the two-thirds of MPS II patients that are cognitively affected. Moreover, severe anaphylactic reactions to the replacement enzyme have been reported (Muenzer *et al*, 2006, 2011), as well as neutralizing antibodies to the enzyme (Scarpa *et al*, 2011), which may decrease efficacy of the treatment (Brooks *et al*, 2003).

Allogeneic stem cell transplantation, although recommended to treat neurological symptoms in MPS I Hurler (Boelens *et al*, 2013; Aldenhoven *et al*, 2015a,b), has had variable outcomes in MPS II. It is also associated with high rates of morbidity and mortality, primarily caused by rejection and graft-vs.-host disease (Vellodi *et al*, 1999; Guffon *et al*, 2009). Recent studies in Japan have described some benefits to the neurological phenotype if stem cell transplant is given at earlier disease stages (Kubaski *et al*, 2017). However, although the level of enzyme delivered from an

Stem Cell and Neurotherapies, Division of Cell Matrix Biology & Regenerative Medicine, Faculty of Biology, Medicine and Health, The University of Manchester, Manchester, UK
*Corresponding author. Tel: +44 161 3060516; E-mail: brian.bigger@manchester.ac.uk

allogeneic transplant is sufficient to clear primary storage material in peripheral organs, it is likely to be the limiting factor for complete neurological correction (Visigalli *et al*, 2010; Biffi & Visigalli, 2013). Indeed, *supra*-physiological enzyme levels achieved after hematopoietic stem cell gene therapy (HSCGT) approaches have been shown to correct neurological disease manifestations in mouse models of MPS I, MPS IIIA, and MPS IIIB (Biffi *et al*, 2004; Visigalli *et al*, 2010; Langford-Smith *et al*, 2012; Sergijenko *et al*, 2013; Holley *et al*, 2017) and in patients with metachromatic leukodystrophy (Sessa *et al*, 2016). Donor myeloid cells from the engrafted cells traffic into the brain and engraft as macrophages taking on similar roles to the resident microglial cell population and deliver enzyme that can be taken up to cross-correct affected neurons (Biffi & Visigalli, 2013; Sergijenko *et al*, 2013; Wilkinson *et al*, 2013).

To date, HSCGT using second-generation lentiviral vectors (LV) has shown limited efficacy in MPS II (Wakabayashi *et al*, 2015), possibly due to insufficient enzyme production from transduced cells. Alternative approaches of direct injection of adeno-associated vectors (AAVs) into the CNS (Cardone *et al*, 2006; Hinderer *et al*, 2016; Motas *et al*, 2016; Laoharawee *et al*, 2017) have yielded promising results in mice and dogs, but outcomes in patients with CLN2, MPS IIIA, or MPS IIIB using this approach have so far been disappointing (Worgall *et al*, 2008; Tardieu *et al*, 2014, 2017) and additionally this does not address the somatic disease. Scale-up from the mouse brain to the human brain is the primary hurdle for direct vector approaches, as are adequate distribution of the therapeutic vector throughout brain tissue and the presence of pre-existing antibodies against AAVs in many patients.

We sought to combine several approaches in HSCGT to optimize the efficiency of this approach to treat whole-body disease in MPS II. Previous studies have exploited the use of LDLR-binding domain peptides, by fusing them to enzymes of interest, and have shown efficient delivery of these chimeric constructs across the BBB in animal models (Spencer & Verma, 2007; Sorrentino *et al*, 2013; Wang *et al*, 2013; Bockenhoff *et al*, 2014; El-Amouri *et al*, 2014; Spencer *et al*, 2014). We took this one step further by combining our myeloid-specific HSCGT approach that we have previously shown to deliver superior levels of enzyme to the CNS in MPS IIIA and MPS IIIB (Sergijenko *et al*, 2013; Holley *et al*, 2017) with increasing the ability of somatic IDS enzyme to cross the BBB. This was achieved by fusing codon-optimized *IDS* to the receptor-binding domain of human apolipoprotein E (ApoE) as a tandem repeat for increased efficacy, by means of an invariant flexible linker at the C-terminal cloned into a third-generation lentiviral vector with the myeloid-specific CD11b promoter, herein referred to as LV.IDS.ApoEII. Using LV.IDS.ApoEII, we observed a complete correction of working memory deficits, neuro-inflammation, and HS storage in the brain, together with normalized rotarod activity, peripheral inflammation, and other somatic disease markers associated with MPS II. These were only partially corrected with the unmodified LV.IDS vector, suggesting IDS.ApoEII provides far superior correction. Importantly, we observed a tripartite mechanism of action for ApoEII: increased active enzyme in plasma, increased uptake, and transcytosis across brain endothelial cells via both ApoE/HS binding receptors and M6P receptors.

# Results

## Development and *in vitro* validation of blood–brain barrier-targeting IDS enzyme

We generated lentiviral vectors encoding for codon-optimized human IDS alone (LV.IDS), or codon-optimized human IDS fused via an invariant flexible linker to a tandem repeat of the human ApoE receptor-binding region (LV.IDS.ApoEII), under the human myeloid-specific CD11b promoter (Fig 1A and B). The CD11b marker is highly expressed by myeloid lineages, both peripherally and in the CNS, with monocytes engrafting in the brain as macrophages with similar functions to microglia.

As the addition of a linker and peptide can alter protein folding and detrimentally affect enzyme activity, we verified that our construct still allowed for IDS overexpression and secretion, and detected 27-fold and 26-fold increases in cellular activity (Fig 1C) and eightfold increases in secreted IDS activity with both LV.IDS and LV.IDS.ApoEII (Fig 1D). Therefore, the C-terminal modifications in the LV.IDS.ApoEII vector do not negatively impact secretion or expression of the modified IDS enzyme *in vitro*.

The primary aim of this study was to assess *ex vivo* HSCGT using an enzyme coupled to a BBB-targeting peptide as a treatment for the cognitive, motor, and skeletal phenotype in the MPS II mouse model. Hence, we transplanted lineage-depleted MPS II hematopoietic stem cells (HSCs) transduced with either LV.IDS or LV.IDS.ApoEII into 16 busulfan-conditioned 6- to 8-week-old MPS II mouse recipients (Fig 1E). We compared this to a normal bone marrow transplant by delivering total bone marrow cells from WT mice into fully myelo-ablated MPS II recipients (WT-HSCT) and compared these to normal WT and MPS II mice as controls. Colony-forming unit (CFU) assays were performed on cells used for transplants, and IDS activity and vector copy number (VCN) were measured in resulting colonies. We obtained mean vector copy numbers of 3.1 and 3.8 in the LV.IDS- and LV.IDS.ApoEII-transduced HSCs (Fig 1F) and overexpression of intracellular IDS enzyme by 124-fold and 152-fold over WT, respectively (Fig 1G). Flow cytometry analysis of peripheral leukocytes at 4 weeks post-transplant demonstrated full engraftment of transduced cells into MPS II recipients, achieving between 80 and 100% of donor CD45.1$^+$ cells (Fig 1H).

## LV.IDS and LV.IDS.ApoEII hematopoietic stem cell gene therapies improve IDS enzyme activity in the brain and express *supra*-physiological levels of active IDS in peripheral organs

To assess therapeutic efficacy of this novel gene therapy in MPS II, mice were sacrificed after behavioral analysis and X-ray imaging at 8 months of age for biochemical analysis of the brain and peripheral organs (*n* = 6). Vector integrations of on average 1–4 were detected in total BM, WBCs, and spleen, together with vector detected in non-hematopoietic organs including the brain, lung, liver, and heart (Fig 2A–D and Appendix Fig S1A–C).

*Supra*-physiological levels of IDS activity were observed: 962 and 613% of WT levels in BM, 115.7 and 354.5% of WT levels in plasma, 413 and 291% of WT levels in spleen, 482 and 384% of WT levels in liver, 324 and 203% of WT levels in heart, and 186 and 113% of WT levels in lung, for LV.IDS and LV.IDS.ApoEII, respectively (Fig 2E–I and Appendix Fig S1D–E). IDS enzyme

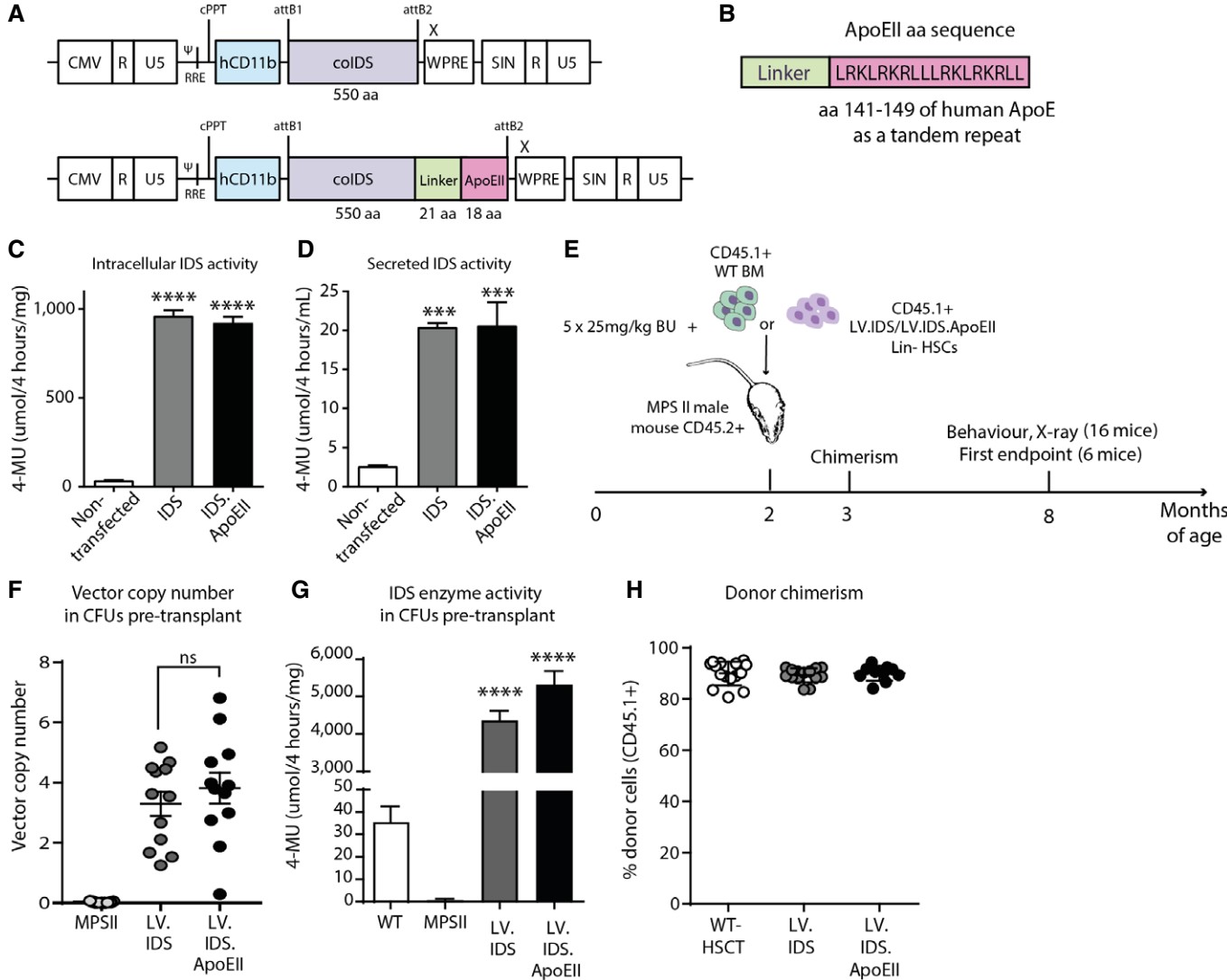

**Figure 1.  Generation and validation of a blood–brain barrier-crossing IDS enzyme.**

A    pCCL lentiviral vectors under the CD11b promoter encoding for the codon-optimized human IDS gene, or the human IDS gene followed by a flexible linker and the ApoEII peptide sequence as a tandem repeat.

B    The invariant linker and ApoEII tandem repeat added at the C-terminal of the IDS gene.

C, D    Intracellular (C) and secreted (D) IDS enzyme activity measured in a human microglial cell line after transfection with 2 μg plasmid DNA of either LV.IDS or LV.IDS.ApoEII for 24 h, and measured 48 h post-transfection, $n = 3$. ***$P < 0.001$, ****$P < 0.0001$ vs. non-transfected.

E    The HSCGT strategy. Busulfan-conditioned 6- to 8-week-old MPS II mice were transplanted with $4 \times 10^5$ Lin⁻ HSCs transduced with LV.IDS or LV.IDS.ApoEII, or $1 \times 10^7$ total bone marrow cells. Six animals per group were sacrificed at 8 months of age for biochemical and microscopic analysis.

F, G    Vector copy number (F) and IDS enzyme activity (G) in transduced BM progenitors from colony-forming unit assays at transplant. $n = 12$ pooled CFUs.

H    Donor chimerism (percentage of CD45.1⁺ cells) in WBCs measured by flow cytometry at 4 weeks post-transplant in transplanted mice.

Data information: In (C–H), data are mean ± SEM, one-way ANOVA, ***$P < 0.001$, ****$P < 0.0001$ vs. MPS II (shown above bars); other comparisons are indicated by brackets.

activity levels in the brain were also elevated compared to untreated MPS II animals (Fig 2F), and are equivalent to 3.4 and 3.7% of normal IDS activity in WT brains for LV.IDS and LV.IDS.ApoEII, respectively. WT-HSCT transplants restored IDS levels equivalent to WT in BM, plasma, and spleen only, whilst no noticeable increase in IDS activity was detected in the brain over untreated MPS II animals (Fig 2F). Although vector copy numbers (VCNs) in all organs measured were generally lower in the LV.IDS.ApoEII

group, the average enzyme activity per VCN (unit/VCN) was similar across all organs, although often trending to slightly higher in the LV.IDS.ApoEII group (Appendix Fig S1F–J) except for plasma, where enzyme activity per VCN in LV.IDS.ApoEII-treated mice was threefold higher than in the LV.IDS-treated group (Fig 2J). These data demonstrate a positive correlation between the number of vector genome integrations and enzyme overexpression (Appendix Fig S1F–J).

**Figure 2. LV.IDS and LV.IDS.ApoEII improve brain-specific IDS activity and express *supra*-physiological levels of active IDS in peripheral organs.**

A–D   Vector copy number measured in (A) bone marrow, (B) brain, (C) spleen, and (D) WBCs, *n* = 6 mice/group.

E–I   IDS enzyme activity levels measured in organs taken at 8 months of age, including (E) bone marrow, (F) brain, (G) plasma, (H) spleen, and (I) heart from control and treated mice, *n* = 6 mice/group.

J   VCN in WBCs vs. enzyme activity in plasma in individual mice.

K–M   Levels of lysosomal enzyme β-hexosaminidase activity in (K) the plasma, (L) spleen, and (M) brain of 8-month-old mice, *n* = 6 mice/group.

Data information: All data are mean ± SEM. One-way ANOVA, $^{ns}P > 0.05$, $*P < 0.05$, $**P < 0.01$, $***P < 0.001$, $****P < 0.0001$ vs. MPS II; other comparisons are indicated by brackets. For IDS enzyme activity, log-transformed data were used for statistical analysis.

**Lysosomal enzyme homeostasis is corrected by LV.IDS.ApoEII**

In disease states such as MPS II, endogenous lysosomal enzymes are often upregulated in an attempt to compensate for the deficiency of another lysosomal enzyme (Motas *et al*, 2016). Elevated levels of the lysosomal hydrolase β-hexosaminidase were detected in the plasma (twofold over WT), spleen (1.53-fold over WT), and brain (twofold over WT) of untreated MPS II animals (Fig 2K–M). Plasma, spleen, and brain levels of β-hexosaminidase were fully normalized back to WT levels in both LV.IDS- and LV.IDS.ApoEII-treated groups (Fig 2K–M). WT-HSCT normalized β-hexosaminidase levels in plasma (Fig 2K and L) and ameliorated levels in the spleen and brain without achieving complete normalization to WT levels (Fig 2M).

**HS in the brain is fully normalized with LV.IDS.ApoEII but not LV.IDS**

MPS II mice accumulate both abnormal HS and DS as undegraded primary substrates in all organs. We have previously shown that clearance of HS from the brain and reductions in neuro-inflammation are critical for correction of behavioral abnormalities in MPS IIIA and MPS IIIB mice (Sergijenko *et al*, 2013; Holley *et al*, 2017), although their roles in neuropathology remain elusive. HS and CS/DS glycosaminoglycans were purified from brain samples and analyzed and quantified by reverse-phase HPLC (Fig 3 and Appendix Fig S2). A sixfold increase in total HS was detected in brains of MPS II mice, and this remains unchanged in mice treated with WT-HSCT (Fig 3A). Mice treated with LV.IDS showed a decrease in total HS accumulation in the brain to approximately threefold of WT levels. Most importantly, LV.IDS.ApoEII-treated mice showed a complete normalization of brain HS levels back to WT levels and significantly lower levels of HS when compared to LV.IDS (Fig 3A).

The composition of HS may also play a role in the pathogenesis of disease. In MPS I, we have previously shown that HS with an excess of 2-*O*-sulfation, present in both mice and patients (Holley *et al*, 2011), is responsible for sequestration of the cytokine CXCL12,

resulting in abnormal HSC migration and engraftment (Watson *et al*, 2014). HS composition analysis in MPS II mice showed that 31.1% of brain HS consisted of the fully sulfated UA(2S)-GlcNS(6S), compared to 12.3% in control WT mice (Fig 3B) with similar increases in UA(2S)-GlcNS and commensurate reductions in UA-GlcNS and non-sulfated UA-GlcNAc. This was partially corrected with WT-HSCT, further corrected with LV.IDS, and completely normalized by LV.IDS.ApoEII.

We also studied CS/DS accumulation and patterning. No significant differences were detectable between WT and MPS II mice in total brain CS/DS levels (Appendix Fig S3A). We detected increases in UA(2S)-GalNAc(4S) from 1.03% in WT mice to 6.5% in MPS II mice, with complete correction obtained in the LV.IDS.ApoEII group (Appendix Fig S3B).

**LV.IDS.ApoEII normalizes lysosomal compartment size in neurons throughout the brain**

We sought to determine the effects of increased IDS enzyme levels on lysosomal enlargement and substrate accumulation in neurons (NeuN) using the lysosomal marker LAMP2. WT animals displayed weak, punctate, and perinuclear red LAMP2 staining that only partially co-localized with green NeuN in the motor cortex (layer V/VI). Untreated MPS II animals displayed strong co-localized staining of NeuN and LAMP2 in the motor cortex, caudate putamen, hippocampus, and amygdala, suggesting a heavy lysosomal burden in neurons and satellite glial cells (Fig 4A). LV.IDS mediated lysosomal compartment reductions in neurons in the cerebral cortex (Fig 4A and B), caudate putamen (Fig 4A), and amygdala (Fig 4A and C), with limited correction in the hippocampus, suggesting only a partial correction of primary substrate accumulation, which strongly correlates with the levels of HS detected in the brain (Fig 3A and B). LV.IDS.ApoEII fully normalized lysosomal compartment in neurons in the cortex, caudate putamen, hippocampus, and amygdala (Fig 4A–C). This can only be mediated by cross-correction of enzyme in affected neurons.

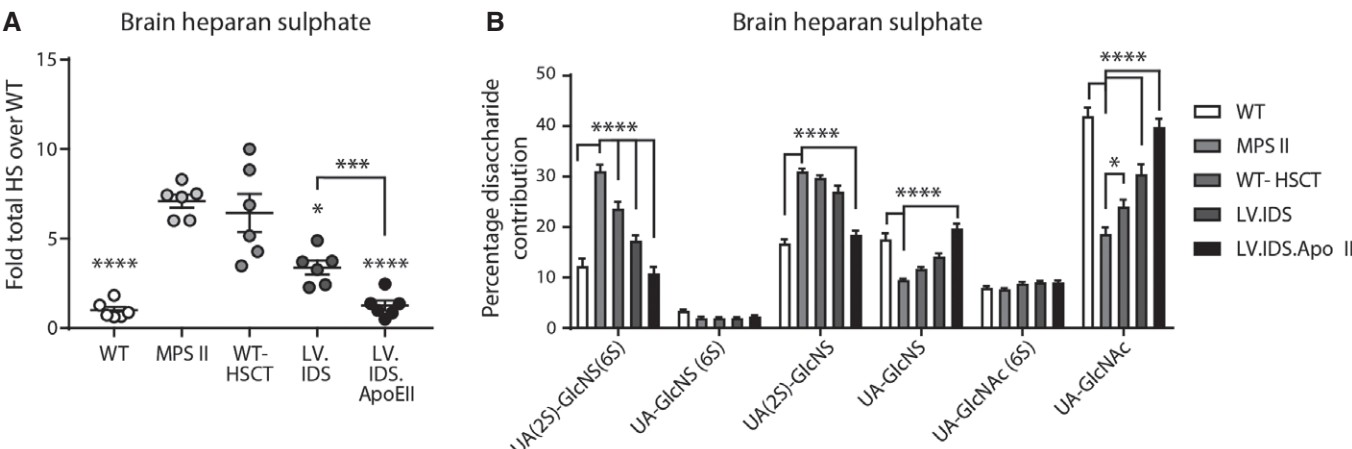

**Figure 3.  Primary accumulation of abnormal HS in the brain is normalized to WT using LV.IDS.ApoEII but only partially by LV.IDS.**

A, B   Total relative amounts of HS (A) and compositional disaccharide analysis of HS (B) from control and treated mice brain samples, *n* = 6 mice/group.

Data information: Data are mean ± SEM, one-way ANOVA (A), and two-way ANOVA (B) with disaccharide and treatments as independent factors, *P < 0.05, ***P < 0.001, ****P < 0.0001 vs. MPS II; other comparisons are indicated by brackets.

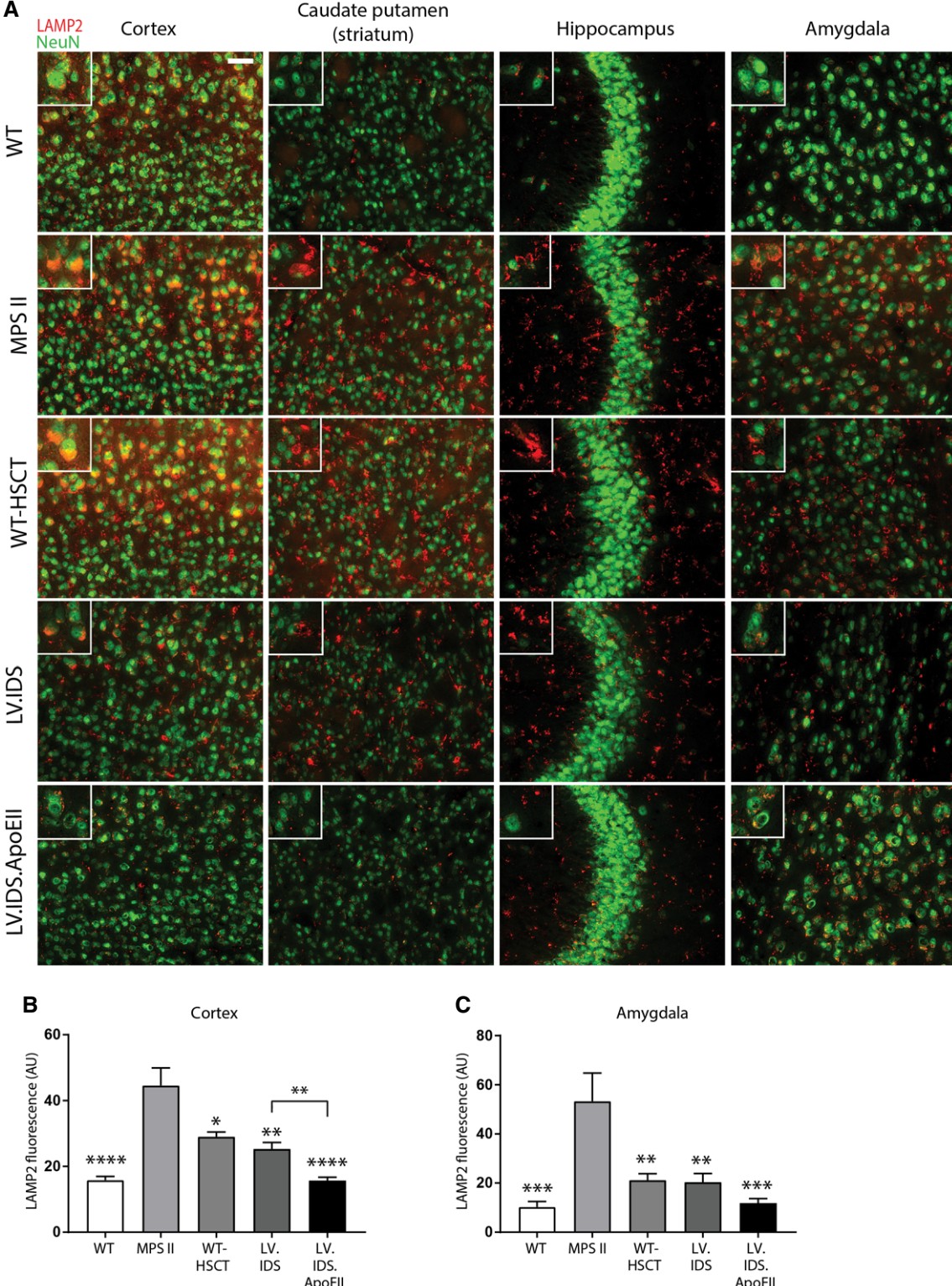

**Figure 4. Lysosomal compartment size is normalized with LV.IDS.ApoEII but only partially with LV.IDS.**

A Representative images of 30-μm brain sections of the motor cortex (M2), caudate putamen (−0.46 mm from bregma), hippocampus (CA3), and amygdala (both −1.22 mm from bregma) from control and treated mice stained with NeuN (green) and LAMP2 (red), *n* = 6 mice/group, 40×; nonlinear adjustments were used equally to reduce background: gamma: 0.72; and input levels: 0–190. Scale bar: 50 μm.

B, C Quantification of LAMP2 immunofluorescence in the cortex (B) and amygdala (C) of 8-month-old MPS II mice, *n* = 3 mice/group. AU: arbitrary units.

Data information: Data are mean ± SEM, one-way ANOVA, *$P < 0.05$, **$P < 0.01$, ***$P < 0.001$, ****$P < 0.0001$ vs. MPS II; other comparisons are indicated by brackets.

**LV.IDS.ApoEII corrects neuro-inflammation in the brains of MPS II mice, whilst LV.IDS mediates improvements**

Astrocytes have been found to mediate a strong neuro-inflammatory response in MPS disorders, which translates into reactive gliosis, astrogliosis, and increased levels of inflammatory cytokines (Wilkinson *et al*, 2012). Brain coronal sections of control and treated MPS II mice were stained with the astrocytic marker GFAP (glial fibrillary associated protein; green) and LAMP2 (red). Significantly more GFAP staining was observed in untreated MPS II than in WT mice in the cortex, caudate putamen, and amygdala, indicative of extensive astrogliosis (Fig 5A). Additionally, strong co-localization of GFAP and LAMP2 was observed in the caudate putamen, hippocampus, and amygdala in untreated and WT-HSCT mice (Fig 5A), suggesting significant lysosomal swelling in astrocytes in addition to neurons (Fig 4A). LV.IDS.ApoEII was able to eliminate astrogliosis and normalize LAMP2 staining to WT levels in the cortex, caudate putamen, hippocampus, and amygdala (Fig 5A–C). LV.IDS partially decreased the number of reactive astrocytes present in the cortex, caudate putamen, and amygdala (Fig 5A–C).

Similar to observations in MPS I, IIIA, and IIIB mice (Wilkinson *et al*, 2012), we also identified significant increases in macrophage inflammatory protein (MIP-1α/CCL3), interleukin-1α (IL-1α) protein, RANTES (CCL5), and monocyte chemoattractant protein (MCP-1/CCL2) in the brains of untreated MPS II animals (Fig 6A–D). Elevated MIP-1α, IL-1α, and RANTES protein levels were fully normalized to WT levels with LV.IDS.ApoEII (Fig 6A–C), suggesting a global reduction in neuro-inflammation and a dampening of the innate inflammatory response. MIP-1α and IL-1α levels were only partially decreased with LV.IDS (Fig 6A and B), but RANTES levels were restored back to WT levels (Fig 6C). However, MCP-1 levels remained elevated for all transplant groups (Fig 6D), probably as a result of busulfan conditioning (Wilkinson *et al*, 2013).

We observed a 30-fold and 25-fold increase in isolectin B4 (ILB4)-positive activated microglial cells in the cortex and striatum of untreated MPS II mice, respectively (Fig 6E–G). Interestingly, both WT-HSCT and LV.IDS reduced ILB4 staining to 14-fold and 12-fold of WT levels in the cortex (Fig 6F), and to 15-fold and 13-fold of WT staining in the striatum (Fig 6G), respectively. Most importantly, LV.IDS.ApoEII treatment completely normalized the number of activated microglia in both the cortex and striatum (Fig 6F and G), which strongly correlates with the reduction of neuro-inflammatory cytokines previously observed and the full abrogation of astrocytosis in LV.IDS.ApoEII. Overall, this suggests that the release of neuro-inflammatory cytokines and chemokines, reactive astrogliosis, and microglial activation in MPS II can be globally abrogated by LV.IDS.ApoEII, and only partially with LV.IDS.

Cognitive evaluation of spatial working memory was evaluated using the Y-maze test, which exploits the mouse's innate preference to explore novel arms over recently explored arms over 10 min and has been used extensively to characterize mouse models of schizophrenia, Alzheimer's disease, and MPS II (Fig 6H; Gleitz *et al*, 2017; Knight *et al*, 2014; O'Leary *et al*, 2014). Neurocognitive assessment showed a decrease in spontaneous alternation between WT and untreated MPS II mice and WT-HSCT-treated mice (Fig 6I). We observed a complete normalization of spontaneous alternation to WT levels in LV.IDS.ApoEII-treated mice, but not in the LV.IDS group (Fig 6I). Importantly, we detected statistically significant differences between LV.IDS and LV.IDS.ApoEII ($P = 0.03$). WT-HSCT had no positive impact on the cognitive symptoms in the Y-maze associated with MPS II. The total number of entries into the different arms of the Y-maze was used as a proxy measure of overall activity to normalize the impact that skeletal abnormalities may have in the MPS II mouse model. No differences in the number of total entries were detected between all tested groups, suggesting a true phenotypic rescue of cognitive symptoms in LV.IDS.ApoEII-treated animals (Appendix Fig S4).

**Somatic disease correction is achieved by all transplant strategies in MPS II mice**

All MPS II patients suffer from severe somatic symptoms including skeletal abnormalities and cardiorespiratory symptoms (Cardone *et al*, 2006; Wraith *et al*, 2008; Gleitz *et al*, 2017). Total body X-rays revealed that the widths of the zygomatic arches (Fig 7A and B), which are enlarged in MPS II mice, were normalized in all transplanted groups. Humerus widths that were thickened in MPS II mice were also significantly reduced in WT-HSCT and LV.IDS groups, although full correction was only obtained in the LV.IDS.ApoEII-treated animals (Fig 7C). Additionally, enlarged femur widths typical of MPS II mice were normalized with LV.IDS and LV.IDS.ApoEII transplant, suggesting significant skeletal rescue in these animals (Fig 7D). To investigate the functional outcomes of therapy, control and treated mice were tested on the rotarod, a well-established test for sensorimotor coordination and balance in movement disorders in rodents (Fig 7E and F; Gleitz *et al*, 2017). MPS II mice showed a reduction in performance on the accelerating rod that was completely rescued by all transplant treatments, including WT-HSCT (Fig 7F).

Global inflammation is typical in MPS II. Thus, we detected significant elevation of cytokines MCP-1, MIP-1α, and RANTES in the livers of MPS II mice at 8 months of age. This was completely abrogated by WT-HSCT, LV.IDS, or LV.IDS.ApoEII treatments (Fig 7G–I), suggesting that all transplant types produce sufficient amounts of peripheral enzyme to correct chronic peripheral inflammation in MPS II livers. We also investigated the expression of *Nppb* and *Myh7*, two markers associated with cardiomyopathies and cardiac pathology, which may be indicators of higher risks of heart failure in the MPS II mouse model. Expression of both *Nppb*, encoding for brain natriuretic peptide (BNP) regulating myocyte stretching and blood pressure, and *Myh7*, encoding for myosin heavy chain beta, a key component of cardiac muscle and type I muscle fibers, was significantly elevated in MPS II mice compared to WT (Fig 7J and K). The expression of both these genes was subsequently normalized to WT levels in the WT-HSCT, LV.IDS, and LV.IDS.ApoEII groups (Fig 7J and K).

**Overexpression of IDS following transplantation of LV.IDS- and LV.IDS.ApoEII-transduced HSCs does not yield an immune response to human IDS**

To study whether gene-modified cells were able to mediate tolerance to human IDS post-transplant, we analyzed plasma from

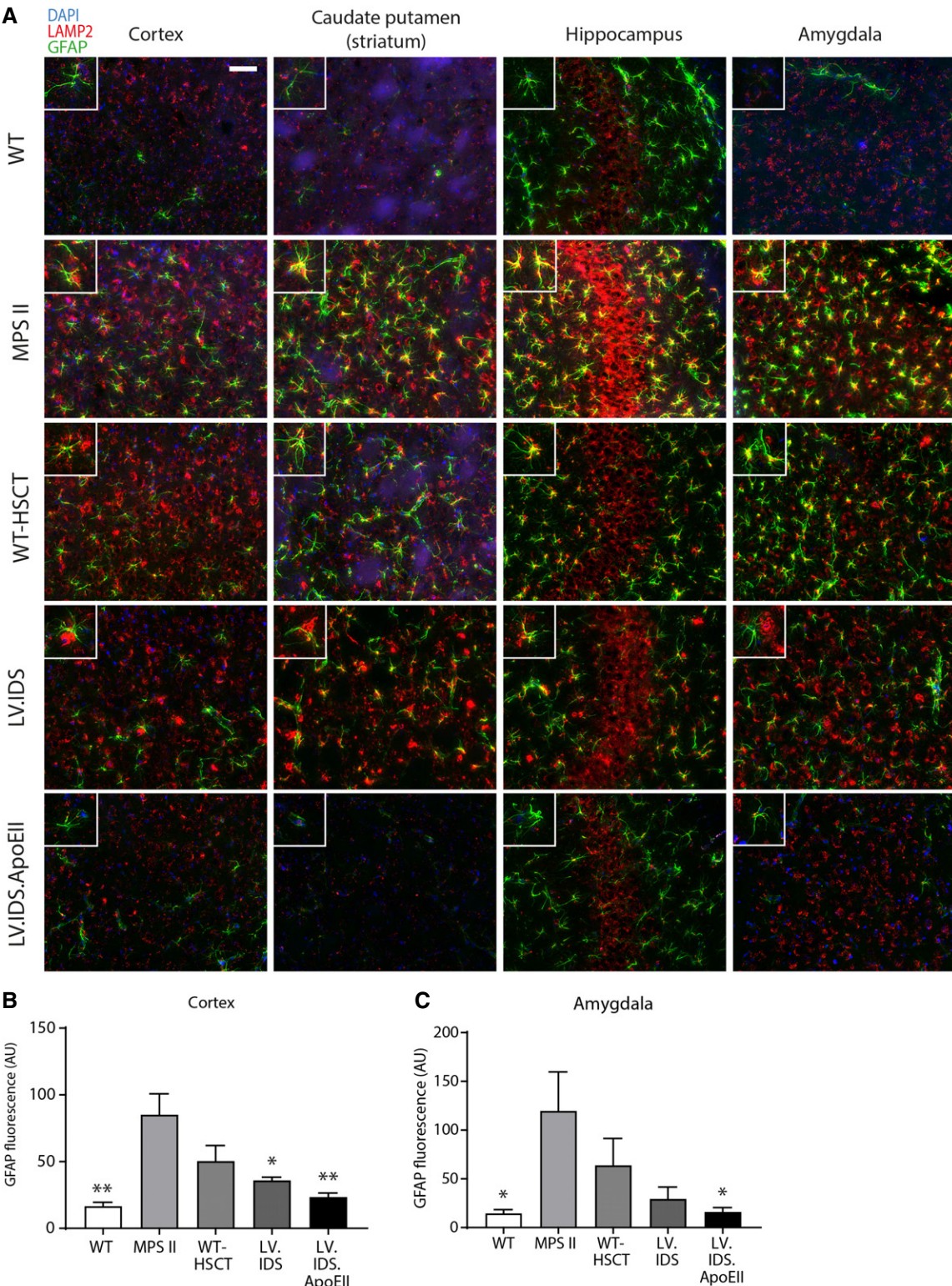

**Figure 5.  LV.IDS.ApoEII normalizes astrogliosis in the brains of 8-month-old MPS II mice.**

A       Representative images of 30-μm brain sections of the motor cortex (M2), caudate putamen (both −0.46 mm from bregma), hippocampus (CA3), and amygdala (both −1.22 mm from bregma) from control and treated mice stained with GFAP (green) and LAMP2 (red), $n$ = 6 mice/group, 40×; nonlinear adjustments were made equally to reduce background: gamma: 0.72; and input levels: 0–190. Scale bar: 50 μm.

B, C    GFAP immunofluorescence was quantified in the cortex (B) and amygdala (C) of 8-month-old MPS II mice, $n$ = 3 mice/group. AU: arbitrary units.

Data information: Data are mean ± SEM, one-way ANOVA, *$P < 0.05$, **$P < 0.01$ vs. MPS II; other comparisons are indicated by brackets.

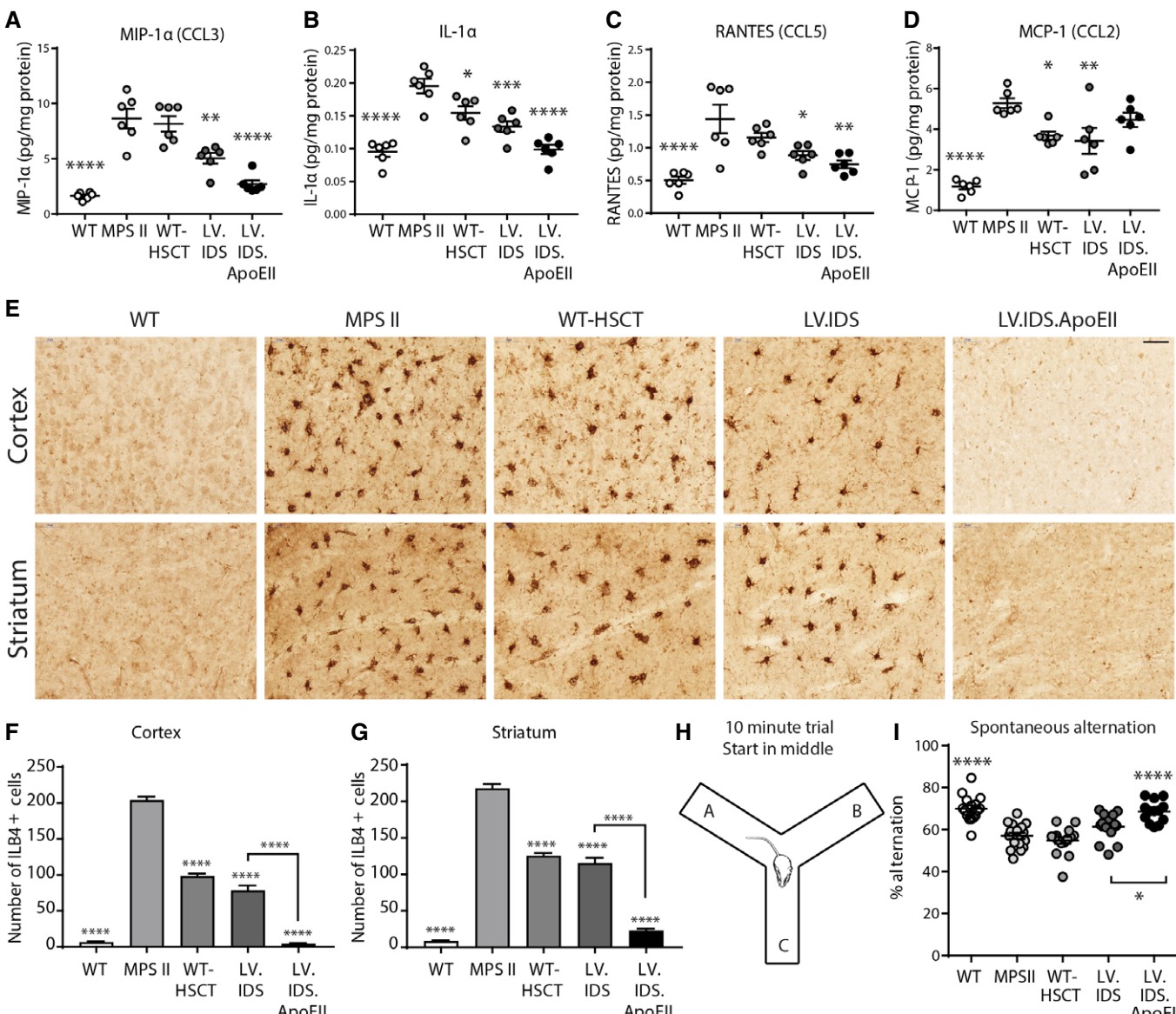

**Figure 6. Neuro-inflammatory cytokines, brain microglial activation, and working memory deficits are normalized using LV.IDS.ApoEII, but not LV.IDS in MPS II mice.**

A–D   Cytokine bead arrays measuring (A) MIP-1α, (B) IL-1α, (C) RANTES, and (D) MCP-1 levels in whole-brain lysate of 8-month-old mice using flow cytometry ($n = 6$ mice/group).

E   Representative images of 30-μm sections of the motor cortex (M2) and striatum stained with isolectin B4 (ILB4) to identify activated microglia, 40×. Scale bar: 50 μm.

F, G   Four 30-μm sections per mouse of the cortex (F) and striatum (G) were counted for the number of ILB4-positive cells (0.26 to −1.94 mm from bregma), $n = 3$ mice/group.

H   Schematic of the Y-maze to measure working memory.

I   Percentage alternation as an indicator of working memory was measured in 10-min trials in the Y-maze at 8 months of age. $n = 12$–$16$ mice per group.

Data information: Data are mean ± SEM, one-way ANOVA, *$P < 0.05$, **$P < 0.01$, ***$P < 0.001$, ****$P < 0.0001$ vs. MPS II; other comparisons are indicated by brackets.

mice that received full myeloablative conditioning followed by either LV.IDS or LV.IDS.ApoEII transplant, both overexpressing human IDS, for IgG antibodies against human IDS. Overall IDS-specific IgG titers in LV.IDS and LV.IDS.ApoEII groups remained in the normal range and did not contribute to an immune response to the enzymes, as expected with this therapeutic approach (Fig 7L).

**LV.IDS.ApoEII treatment acts through multiple mechanisms**

We have previously established that bone marrow-derived monocytes traffic into the brain and engraft as macrophages resembling microglia (Wilkinson *et al*, 2013), releasing enzyme within the brain that can cross-correct neurons (Sergijenko *et al*, 2013; Holley *et al*,

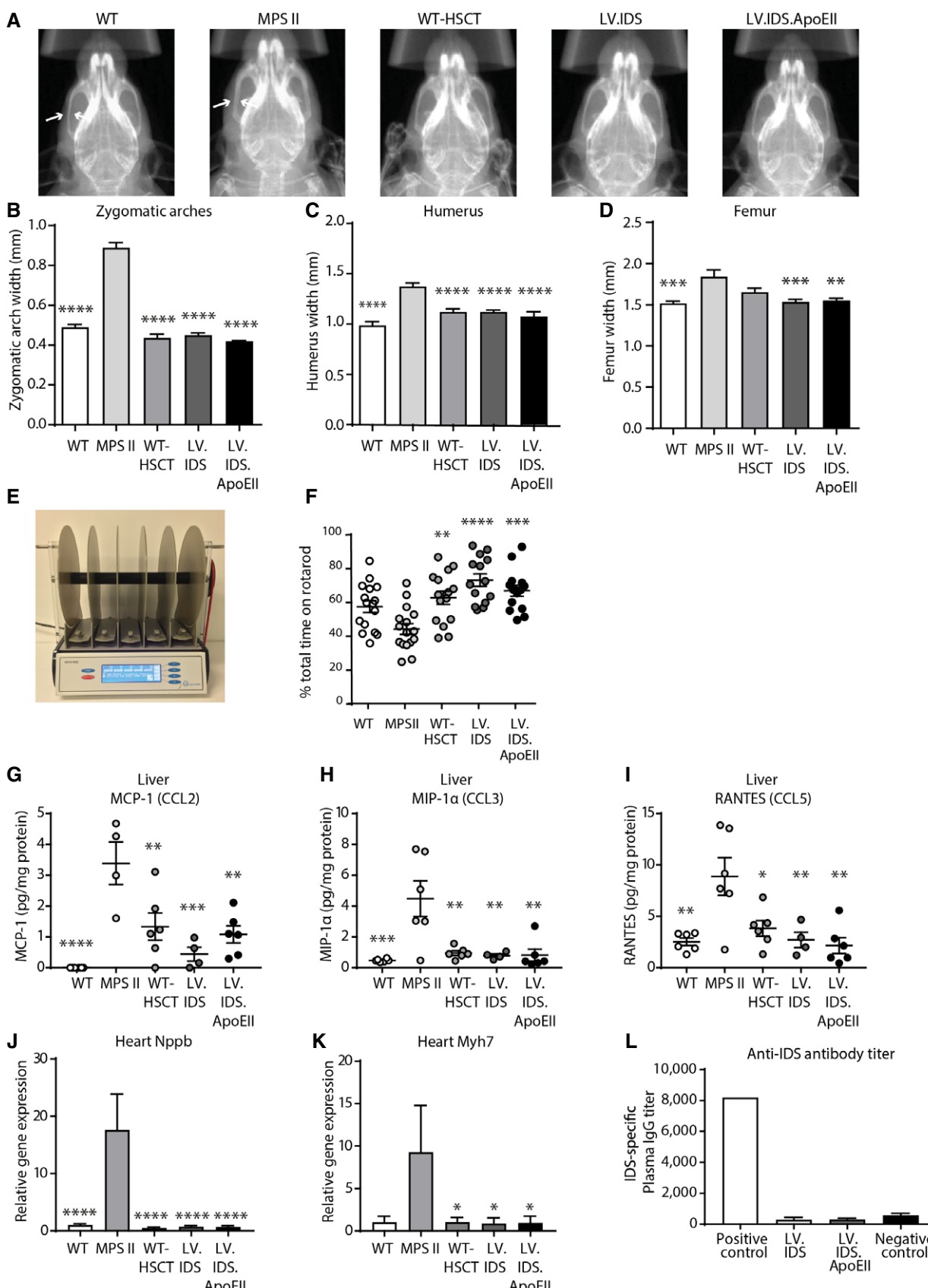

Figure 7.

**Figure 7.   WT-HSCT, LV.IDS, and LV.IDS.ApoEII normalize skeletal defects, peripheral inflammation, and heart failure markers in MPS II mice.**

A      Representative X-ray images of control and treated MPS II mouse craniums at 8 months of age. White arrows indicate the area where zygomatic arches are measured.
B–D    Zygomatic arch widths (B), humerus widths (C), and femur widths (D) were analyzed using ImageJ software, *n* = 5–11.
E      Representation of the accelerating rotarod.
F      Eight-month-old control and treated mice were trialed three times on the accelerating rotarod for a maximum of 300 s (4–4 rpm over 300 s), *n* = 12–16 mice/group.
G–I    Cytokine bead arrays measuring MCP-1 (G), MIP-1α (H), and RANTES (I) were performed on liver lysates of 8-month-old mice using flow cytometry, *n* = 4–6.
J, K    Gene expression of *Nppb* (J) and *Myh7* (K) in hearts of control and treated MPS II animals (WT *n* = 3; MPS II *n* = 4; WT-HSCT, LV.IDS, and LV.IDS.ApoEII *n* = 6).
L      IDS-specific IgG antibody titers were measured in plasma (pos. control *n* = 1, neg. control (uninjected MPS II plasma) *n* = 6, LV.IDS/LV.IDS.ApoEII *n* = 6).

Data information: Data are mean ± SEM, one-way ANOVA, *P < 0.05, **P < 0.01, ***P < 0.001, ****P < 0.0001 vs. MPS II.

2017). This is the primary mechanism of action for the LV.IDS treatment. The addition of the ApoEII peptide to IDS may change this interaction in several ways, including improving blood–brain barrier uptake, modifying the stability or activity of the enzyme, and changing its uptake/cell association properties.

Following the observed increase in enzyme activity in plasma and the increased enzyme activity per VCN in LV.IDS.ApoEII-treated mice (Fig 2G and J), we hypothesized that the relative activity of the enzyme might be increased with ApoEII. IDS and IDS.ApoEII protein was produced in CHME3 cells, observing similar levels of enzyme activity per unit of IDS protein between IDS and IDS.ApoEII *in vitro*, both intracellularly and extracellularly (Appendix Fig S5). This suggested that there was not a specific increase in intrinsic enzyme activity, at least *in vitro*. We also hypothesized that circulation time might be increased; thus, we injected equivalent levels of IDS and IDS.ApoEII enzyme into MPS II mice and monitored protein levels by ELISA (Fig 8A). Interestingly, enzyme clearance rate, whether from degradation or uptake into organs, was similar between the IDS and IDS.ApoEII groups and was effectively cleared in both groups by 30 min.

We next compared levels of active enzyme in the plasma compared to total IDS protein measured by ELISA. IDS.ApoEII appeared to be more active per unit of IDS protein in plasma (Fig 8B). It is unclear whether this effect is mediated by a stabilization effect of ApoEII on IDS enzyme activity in plasma or an increase in IDS activity when produced *in vivo*. Notably, this is quite close to (but above) the limit of detection of the activity assay.

We used the BBB endothelial cell line bEND.3 to determine whether there was any difference in enzyme uptake by endothelial cells. These cells produce an effective BBB layer in transwells (Fig 8C) and express both LDLR and LDLR-related protein 1 (LRP1)

receptors (Fig 8D and E). Active IDS and IDS.ApoEII enzyme were produced following transfection of CHME3 human microglial cells, and relative enzyme levels secreted into the culture media were standardized by ELISA (Appendix Fig S5). In monolayer culture, we compared the uptake of IDS vs. IDS.ApoEII into bEND.3 cells, identifying a 4.7-fold increase in cellular uptake with the addition of the ApoEII peptide (Fig 8F). We subsequently compared uptake and transcytosis of IDS or IDS.ApoEII across polarized cell layers of bEND.3 endothelial cells in transwells to mimic the BBB. 2.6-fold increases in uptake were apparent, similar to when cells were grown in monolayer, together with a 1.5-fold increase in transcytosis to the basolateral side of the transwell (Fig 8G and H).

To determine the mechanism of uptake, blockade of ApoE-dependent receptors using ApoE or blockade of M6P receptors using M6P (Fig 8I) revealed that IDS.ApoEII was preferentially and unexpectedly taken up *via* M6P receptors, although blockade with ApoE peptide also significantly inhibited uptake of IDS.ApoEII by approximately 30%. The ApoEII peptide sequence also codes for a HS site, required as a co-receptor for receptor/ligand binding of ApoE to its receptors, including LDLR and LRP1 (Ji *et al*, 1993). Heparinase pretreatment of cells prior to protein addition also resulted in a significant reduction in IDS.ApoEII uptake by approximately 30%, similar to results with ApoE peptide, suggesting that ApoE/HS may also be important in uptake and transcytosis across bEND.3 cells. Smaller decreases with ApoEII and HS treatment were also seen when IDS protein was added, but these were not significant.

## Discussion

We demonstrate complete correction of working memory deficits, neuro-inflammation, and HS storage, together with rotarod activity,

**Figure 8.   IDS.ApoEII shows increased plasma stability and enhanced uptake by brain endothelial cells via multiple mechanisms.**

A      Plasma clearance of IDS protein measured in MPS II mice injected with 12 ng of total IDS enzyme at 1, 10, and 30 min post-injection, *n* = 2/group.
B      Correlation between plasma IDS enzyme activity and IDS protein measured by ELISA.
C      FITC–dextran uptake into bEND.3 cells (*n* = 6).
D, E    Immunofluorescent staining for LDLR (D) or LRP1 (E) in bEND.3 cells. Scale bar: 50 μm.
F      Uptake of IDS or IDS.ApoEII produced by CHME3 cells added to growth media of bEND.3 cells grown in monolayer culture for 24 h. *n* = 2 independent experiments, with 3 wells/condition.
G      Uptake of IDS or IDS.ApoEII produced by CHME3 cells added to growth media of bEND.3 cells grown in transwell culture for 24 h. *n* = 3 independent experiments, with 2 wells/condition.
H      Percentage transcytosis to the basal layer of bEND.3 cells from (G).
I      Receptor-mediated uptake of IDS or IDS.ApoEII after 24 h, following heparinase treatment of the cell layer or blocking with 7.5 mM M6P or 50 μg/ml human recombinant ApoE for 1 h prior to enzyme addition. *n* = 3 wells/condition.

Data information: Data are mean ± SEM, one-way ANOVA or *t*-test as appropriate, *P < 0.05, **P < 0.01, ***P < 0.001, ****P < 0.0001. For panel (I), significance values above bars are vs. IDS control; all other comparisons are indicated by brackets.

peripheral inflammation, and other somatic disease markers associated with MPS II by HSCGT using an IDS.ApoEII-tagged enzyme. Significantly, IDS.ApoEII provides superior correction to native IDS

enzyme, where only partial neurological correction was achieved. As most patients with MPS II have the severe form of the disease with neurological involvement, this is the most important aspect of

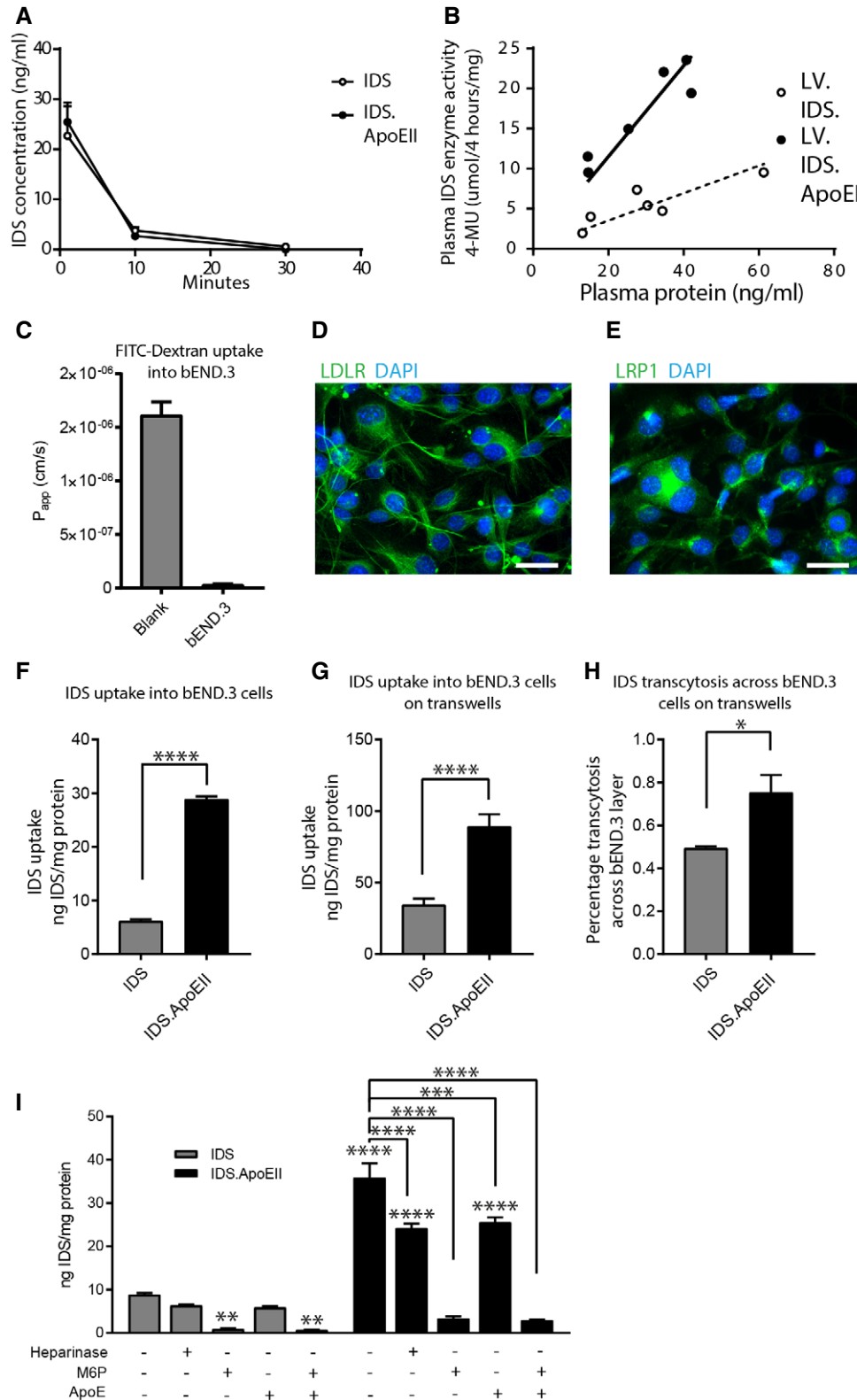

Figure 8.

disease to target, with current therapies failing to treat the brain. Thus, IDS.ApoEII provides a potential therapy for MPS II patients. We previously showed that lentiviral vectors under the myeloid-specific CD11b promoter were able to mediate neurological correction in mouse models of MPS IIIA (Sergijenko *et al*, 2013) and MPS IIIB (Holley *et al*, 2017), following delivery of the missing enzyme via HSCGT. Brain correction is mediated by monocyte trafficking to the brain and engraftment of these cells as macrophages with similar functions to microglia (Wilkinson *et al*, 2013). Enzyme released by these cells is believed to mediate cross-correction of neurons (Capotondo *et al*, 2012; Sergijenko *et al*, 2013). Thus, the fact that native IDS was unable to fully correct brain manifestations despite its efficacy at correcting somatic manifestations of MPS II was surprising. MPS II mice show gait abnormalities from early in life and progressively reducing performance on the rotarod (Gleitz *et al*, 2017). In our hands, WT-HSCT was only successful at correcting somatic disease and rotarod function, a phenotype that we regard as being primarily associated with the skeletal dysfunction in MPS II mice. This is also in keeping with HSCT outcomes in patients that although there may be some benefit when given very early in life, there are little data yet to support its widespread use in patients with the severe form of the disease (Vellodi *et al*, 1999; Guffon *et al*, 2009; Kubaski *et al*, 2017).

The efficacy of a previous HSCGT approach in MPS II has been difficult to evaluate, primarily due to the methods used not being sensitive enough to accurately determine phenotypic correction; Wakabayashi *et al* (2015) used semi-quantitative vital dye histopathology together with glycosaminoglycan dye binding assays, which have a poor limit of detection in the brain and measure all charged carbohydrates, not just glycosaminoglycans. Enzyme levels in the brain obtained in this study were slightly lower than we obtained with LV.IDS (2.9% compared to 3.4 and 3.7% obtained with LV.IDS and LV.IDS.ApoEII, respectively), but this was also not compared to WT-HSCT.

Previous gene therapy studies using CNS or cerebroventricular injections of AAV9 vectors in MPS II mice have shown increases up to 45% of normal IDS activity levels in the brain, with significant reductions in total glycosaminoglycan burden (Hinderer *et al*, 2016; Motas *et al*, 2016; Laoharawee *et al*, 2017). Here, we detected only moderately increased IDS levels in brain to 3.7% of WT with IDS.ApoEII; however, we were able to completely clear HS and DS storage in the brain, together with achieving *supra*-physiological levels of IDS enzyme in BM, liver, spleen, liver, heart, and lung, effectively correcting somatic disease. Although the AAV approach may have merit in the clinic, it has not yet demonstrated efficacy in the brain, with disappointing clinical outcomes to date in CLN2, MPS IIIA, and MPS IIIB, despite good preclinical data (Worgall *et al*, 2008; Tardieu *et al*, 2014, 2017). Several factors limit this approach: notably scale-up and distribution in the brain and antibodies against AAV or delivered enzyme. Our HSCGT approach is not limited by scale-up as has previously been shown for metachromatic leukodystrophy, with 10% brain enzyme in the mouse model (Biffi *et al*, 2004) and twofold normal CSF enzyme achieved in patients following treatment (Biffi & Visigalli, 2013; Sessa *et al*, 2016). We have also shown here that the HSCGT approach tolerizes animals to the delivered enzyme. Immunologically foreign proteins and enzymes such as ERT can trigger the release of inhibitory antibodies that may decrease therapeutic efficacy. IgG antibodies against human

recombinant IDS in plasma of LV.IDS- and LV.IDS.ApoEII-treated mice were undetectable, suggesting that HSCGT generates immune tolerance as previously shown (Bracy *et al*, 2001; Bigger *et al*, 2006; Visigalli *et al*, 2010).

Importantly, the addition of the linker and ApoEII peptide did not alter the expression, production, secretion, or activity of the IDS enzyme *in vitro*, but produced *supra*-physiological levels of IDS enzyme in transduced HSCs using both LV.IDS and LV.IDS.ApoEII vectors with similar vector copy number *in vitro* and *in vivo*. Surprisingly, IDS enzyme activity levels in plasma in the LV.IDS.ApoEII group were approximately threefold higher than in LV.IDS, even with lower VCN in WBCs, which translates to higher enzyme activity per copy. It is unclear whether this effect is due to increased enzyme stabilization in plasma or increased specific activity of IDS enzyme, by the addition of the ApoEII peptide. Neither effect is evident *in vitro*. We also observed significantly increased uptake of IDS.ApoEII in bEND.3 cells compared to IDS both *via* an ApoE/HS-based mechanism and *via* M6P receptors. The use of multiple targeting mechanisms could mediate more efficient targeting to enzyme-deficient cells throughout the body as well as across the BBB. The receptor-binding portion of ApoE used here is able to form a high-affinity binding complex with an octasaccharide HS fragment composed of four repeats of UA(2S)-GlcNS(6S) (Libeu *et al*, 2001), which are abundant on endothelial cell surfaces and even more abundant in MPS II. In the context of this study, increases in sulfated disaccharides observed in diseased animals could promote improved cellular uptake of the IDS.ApoEII enzyme, *via* the HS binding motif in the ApoEII peptide, thereby enhancing enzyme targeting to diseased cells. HS typically acts as a co-receptor in many receptor–ligand interactions, and increased binding to HS proteoglycans through ApoEII could mediate an increase in cellular uptake through the LDLR, LRP1, or M6P or by direct uptake of an ApoE–HSPG complex (Mahley & Huang, 1999; Libeu *et al*, 2001). Increased cellular association, coupled with more effective enzyme internalization into BBB and other cells, could together account for the normalization of HS and DS storage, as well as several other neuropathologies in the brain seen with LV.IDS.ApoEII, with the same overall levels of enzyme in the brain. Enzyme turnover within cells could account for the similarities in enzyme activity observed in the brain. A similar effect of increased uptake, cell association, and clearance of storage material for similar enzyme activities has been seen in MPS IIIB (Kan *et al*, 2014) and Pompe disease (Maga *et al*, 2013) using enzymes coupled to GILT tags targeting an alternative epitope of M6P *via* an IGFII receptor-directed peptide. Our glycosaminoglycan clearance results in the brain are similar to a study delivering AAV expressing SGSH.ApoB in MPS IIIA mice (Sorrentino *et al*, 2013). Although in this study they were able to show small increases in brain enzyme activity with SGSH.ApoB, here we saw no significant changes over native IDS.

Neuro-inflammation is commonly reported in LSDs, likely caused by the accumulation of various undegraded molecules, which cooperatively activate and perpetuate a neuro-inflammatory milieu that may exacerbate the disease itself. MPS II mice elicit a strong inflammatory response in the brain, with normalization in the LV.IDS.ApoEII group, with only partial decreases in LV.IDS-treated mice. Microglial activation and astrocytosis are commonly reported in MPS disorders, including this study (Ohmi *et al*, 2003; Villani *et al*, 2007; Wilkinson *et al*, 2012; Martins *et al*,

2015). We observed a complete abrogation of activated microglia and astrocytes, together with restoration of lysosomal compartment size in the brain of LV.IDS.ApoEII mice, with only partial correction in the LV.IDS group. The correction achieved with LV.IDS.ApoEII is comparable to the correction of astrogliosis and microglial activation reported using direct AAV9-IDS injection into the CSF (Motas *et al*, 2016). Additionally, we observe correction of peripheral inflammation in all transplant groups, indicating that peripheral IDS enzyme levels obtained with an allogeneic transplant can mediate a reduction in inflammation in the periphery.

We recently determined that spatial working memory, coordination, and balance were significantly impacted in the MPS II mouse model at 8 months of age (Gleitz *et al*, 2017) and corresponded to the progressive phenotype of neurodegeneration and loss of mobility observed in MPS II patients. As a sensitive and widely accepted paradigm of exploratory behavior and spatial working memory, the Y-maze accounts for potential physical impairments in MPS II mice, unlike the Barnes maze, which may be invalidated by differential physical performance (Laoharawee *et al*, 2017). Full behavioral correction of cognitive deficits was observed in the LV.IDS.ApoEII group alongside normalization of coordination and balance. We hypothesize that cognitive improvements likely stem from a combination of factors: a reduction in primary storage of HS alongside full abrogation of chronic neuro-inflammation, astrogliosis, and microglial activation, all of which were only observed in LV.IDS.ApoEII-treated animals. The rescue of coordination and balance can be attributed to either central or peripheral rescue, or a combination thereof. Most importantly, this further highlights that the addition of the ApoE tandem peptide to IDS is required to provide a full correction of the neurocognitive aspect in MPS II mice, although a much higher VCN in LV.IDS-treated mice might achieve the same effect.

We and others have reported progressive skeletal abnormalities in the MPS II mouse model, such as enlargement of craniofacial bone structures and femurs (Cardone *et al*, 2006; Gleitz *et al*, 2017), correlating with the dysostosis multiplex seen in MPS II patients. In patients, ERT using idursulfase shows limited benefits in joint pain, stiffness, or range of motion, although earlier treatments could provide benefits (Tomatsu *et al*, 2015). In our study, the widths of zygomatic arches, humerus, and femurs were significantly reduced in all transplanted animals, including WT-HSCT, suggesting that some level of enzyme can penetrate bone tissue if treated at an early time point when skeletal phenotype remains mild. This is partly comparable to liver-directed AAV2/8TBG-IDS gene therapy, where craniofacial abnormalities were also corrected (Cardone *et al*, 2006). Importantly, it is likely that sustained availability of enzyme to the skeleton and joints from an early time point is required for clinical improvements.

Significantly, the addition of the ApoE tandem repeat to human IDS is the vital component that allows for a comprehensive correction of neuro-inflammatory markers, lysosomal storage, and cognitive working memory in the MPS II mouse model, likely by a combination of increased uptake across the BBB via ApoE-dependent mechanisms and into all cells via M6P receptors, although increased plasma activity cannot be discounted. This is the first study highlighting the combined use of HSCGT and the ApoEII-fusion enzyme to effectively correct many aspects of the neurological, skeletal,

inflammatory, and behavioral phenotypes in MPS II mice. We present here a strong proof-of-principle for the clinical development of HSCGT using ApoEII-modified enzyme for the treatment of MPS II patients or other enzyme deficiencies that impact the brain.

# Materials and Methods

### Expression vectors

Human IDS cDNA was codon-optimized and synthesized by GeneArt (ThermoFisher, Paisley, UK) and cloned into the third-generation LV pCCL.sin.cPPT.hCD11b.ccdB.wpre to create pCCL.sin.cPPT.hCD11b.IDS.wpre. The brain-targeting peptide sequence ApoEII as a tandem repeat (LRKLRKRLLLRKLRKRLL) was inserted downstream of the codon-optimized human IDS cDNA using the long invariant linker (LGGGGSGGGGSGGGGSGGGGS; Bockenhoff *et al*, 2014), synthesized using GeneArt, and cloned into a third-generation lentiviral backbone to create pCCL.sin.cPPT. hCD11b.IDS.ApoEII.wpre.

### Transfection, uptake, and transcytosis

Human microglial CHME3 cells were transfected with 2 μg of CD11b.IDS or CD11b.IDS.ApoEII plasmid DNA using 7.5 mM high-potency linear polyethylenimine (pH 7.4, 40 kDa; Polysciences Inc., Warrington, PA, USA) and 150 mM NaCl. Media/cell lysates were collected after 48 h. Human IDS protein was detected using the Human Iduronate-2-Sulfatase DuoSet ELISA kit according to the manufacturer's instructions (R&D Systems, Abingdon, UK). For receptor-mediated studies, 70–140 ng of either IDS or IDS.ApoEII was added to bEND.3 cells following seeding at 10,000 cells/cm$^2$ for 3 days for monolayer culture or 50,000 cells/cm$^2$ for 5 days on the luminal side of Thincerts (0.4 μm pore size, translucent PET). Cell lysate and media were collected after 3 h. For blocking experiments 7.5 mM M6P (Sigma), 50 μg/ml of recombinant human apoE (Abcam), or 0.4 mIU heparinase I, II, and III (GrampEnz) was added to bEND.3 cells for 1 h prior to the addition of IDS or IDS.ApoEII for 3 h. FITC–dextran (mW 70 kDa, 4.7 mg/ml) was added for 3 h prior to measurement of fluorescence (excitation 435 nm, emission 490 nm) and P$_{app}$ calculation.

### LV production and titration

Lentiviral vector was produced as previously described (Sergijenko *et al*, 2013) by transient transfection of HEK293T cells with pRSV-Rev, pMDLg/pRRE, pMD2.G (Didier Trono, Addgene plasmids), and LV expression plasmid (Siapati *et al*, 2005; Bigger *et al*, 2006; Langford-Smith *et al*, 2012; Sergijenko *et al*, 2013) and 7.5 mM polyethylenimine (40 kDa; Polysciences; Kuroda *et al*, 2009). EL4 mouse lymphoma cells (ATCC TIB-39; ATCC, Manassas, VA, USA) were transduced with three dilutions of concentrated LV and collected 72 h later. The number of integrated viral genomes per cell, measured using a primer and probe set against *WPRE*, was determined by qPCR using a standard curve generated by dilutions of genomic DNA from an EL4 cell line containing 2 integrated copies/cell of pHRsin.SFFV.eGFP.att.wpre (Langford-Smith *et al*, 2012).

## Mice and transplant procedures

Female mice heterozygous for the X-linked allele on a C57BL/6 background were obtained from Prof. Joseph Muenzer (University of North Carolina at Chapel Hill, NC, USA) and bred with wild-type C57BL/6J males (Envigo, Alconbury, UK). MPS II mice were back-crossed onto the Pep3 CD45.1 congenic background (B6.SJL-*Ptprc*[a]-*Pepc*[b]/BoyJ) to distinguish donor and recipient cells as previously described (Langford-Smith *et al*, 2012). WT littermates were used as controls throughout. Mice were housed in individually ventilated cages with *ad libitum* access to food and water and were kept in a 12-h light/dark cycle. Male mice were used in this study and housed in groups of 2–5 with littermates.

For pharmacokinetic studies, age-matched 12- to 16-week-old MPS II male mice were injected with 200 μl of media containing 60 ng/ml of IDS or IDS.ApoEII enzyme. Blood was obtained at 1, 10, and 30 min post-injection and mixed with citrate buffer (Sigma). Human IDS protein was detected by ELISA as above.

For transplantation studies, total bone marrow mononuclear cells from 6- to 12-week-old male MPS II mice (CD45.1[+]) were isolated from femurs and tibias and lineage-depleted using the murine lineage cell depletion kit (Miltenyi Biotec, Bisley, UK) as previously described (Sergijenko *et al*, 2013). Cells were stimulated using 100 ng/ml murine stem cell factor, 100 ng/ml murine fms-like tyrosine kinase-3, and 10 ng/ml recombinant murine interleukin-3 (Peprotech, Rocky Hill, NJ, USA) for 3 h prior to transduction with a lentiviral vector for 24 h at a MOI of 100.

Six- to eight-week-old male MPS II mice housed in groups in individually ventilated cages were myelo-ablated using 125 mg/kg busulfan (Busilvex; Pierre Fabre, Boulogne, France) in five daily doses (25 mg/kg/day) via i.p. injection. Within 24 h of myeloablation, mice received $3–4 \times 10^5$ lineage-depleted transduced HSCs through the lateral tail vein. For wild-type transplants (WT-HSCT), busulfan-conditioned MPS II mice received $1–2 \times 10^7$ untransduced total bone marrow cells from 6- to 12-week-old WT donors (CD45.1[+]). We recorded three transplant-related deaths (occurring < 5 days post-transplant due to anemia), but no other adverse effects were detected after engraftment.

## Chimerism analysis using flow cytometry

Engraftment of donor HSCs was assessed at 4 weeks post-transplant in peripheral blood, by staining leukocytes with anti-mouse CD45.1-PE (donor leukocytes), anti-mouse CD45.2-FITC (recipient leukocytes), anti-mouse CD3-Pe-Cy5, anti-mouse CD19-APC-Cy7, and anti-mouse CD11b-Pe-Cy7 (BD Biosciences, Oxford, UK) in 5 μM ToPro3 Iodide (ThermoFisher). Analysis was performed on a BD FACSCanto II flow cytometer (BD Biosciences).

## Behavioral analysis

At 8 months of age, the rotarod test was used to evaluate motor coordination and balance as previously described (Gleitz *et al*, 2017). Latency to fall was recorded for all test trials and was calculated as percentage of total trial time. The average of the three trials is shown for each mouse. Spatial working memory was assessed in all mice at 8 months of age in a single, 10-min trial in a Y-maze consisting of three identical arms (O'Tuathaigh *et al*, 2007; Jiang *et al*, 2015;

Gleitz *et al*, 2017). The effect was calculated as percent alternation = [no. of alternations/(total number of arm entries – 2)] × 100.

## X-ray imaging of live mice

Control and treated male mice at 8 months of age were anesthetized using isoflurane (induction: 3 l/min in pure $O_2$; maintenance: 1.5 l/min in pure $O_2$) and radiographed (45 keV) using the Bruker In-Vivo Xtreme system. Due to cost constraints and obvious skeletal changes, only 5–11 mice per group were x-rayed. X-ray images were analyzed using ImageJ software for individual bones widths.

## Sample processing

At 8 months of age, six mice per group were anesthetized and transcardially perfused with 37°C PBS. One brain hemisphere was fixed in 4% paraformaldehyde for 24 h, and transferred to 30% sucrose and 2 mmol/l $MgCl_2$/phosphate-buffered saline solution for 48 h before freezing at −80°C. Samples of brain, spleen, heart, and liver were snap-frozen on dry ice. Bone marrow samples were collected by flushing one tibia and femur with 1 ml 2% FBS/PBS, filtered using a 70-μm cell strainer and lysed using red blood cell lysis buffer (150 mM $NH_4Cl$, 10 mM $KHCO_3$, 0.1 mM EDTA, pH 7.2–7.4). For enzyme activity assays, samples were homogenized and sonicated in homogenization buffer (0.5 M NaCl, 0.02 M Tris, 0.1% Triton X-100, pH 7). Genomic DNA used for organ VCN analysis was extracted using GenElute Mammalian Genomic DNA Miniprep kit (Sigma).

## Enzyme activity assays

IDS enzyme activity was measured in a two-step protocol using the fluorescent substrate MU-αIdoA-2S (Carbosynth) and Aldurazyme (Genzyme) as previously described (Lu *et al*, 2010). Starting material was standardized to 20 μg of total protein or plasma, 40 μg for liver, heart, lung, spleen, and bone marrow, and 60 μg for brain using a BCA assay (ThermoFisher). For β-hexosaminidase activity, 1 μg of total protein from brain, or 2 μg from spleen and plasma were added to 0.5 mM 4-methylumbelliferyl-*N*-acetyl-β-D-glucosaminide substrate (Sigma), incubated for 40 min at 37°C, and stopped with 200 μl of 0.2 M carbonate buffer. Fluorescence was measured using the BioTek Synergy HT plate reader (excitation: 360 nm; emission: 460 nm).

## Immunohistochemistry

Free-floating IHC was performed on 30-μm PFA-fixed coronal brain sections using rabbit anti-NeuN (1:1,000, ab177487; Abcam), rabbit anti-GFAP (1:1,500, Z0334; Dako, Stockport, UK), and rat anti-LAMP2 (1:500, ab13524; Abcam) primary antibodies using standard protocols (Malinowska *et al*, 2010). Isolectin B4 (ILB4, 5 μg/ml, L5391; Sigma) was visualized on 30-μm coronal brain sections using DAB substrate for 40 s (Vector, Peterborough, UK) using standard protocols (Wilkinson *et al*, 2013). bEND.3 cells were fixed in 4% paraformaldehyde and stained using rabbit anti-LDLR and rabbit anti-LRP1 (both Abcam). Images were acquired on a 3DHISTECH Pannoramic-250 microscope slide-scanner using a 20×/0.30 Plan Achromat objective (Zeiss) with extended focus and

the DAPI, FITC, and TRITC filter sets. Snapshots of the slide-scans were taken using CaseViewer software (3DHISTECH). Nonlinear adjustments were made to all immunofluorescence images equally to eliminate background: gamma: 0.72; and input levels: 0–190. GFAP immunofluorescence was quantified using ImageJ software on four sections per mouse for cortex, and one section per mouse for the amygdala ($n = 3$/group). Counts of isolectin B4-positive cells were performed on four sections per mouse ($n = 3$ mice/group) at 20× magnification and counted manually using ImageJ software.

## Glycosaminoglycan analysis

Soluble brain fractions were collected and processed as previously described (Holley *et al*, 2011). HS chains were digested using 0.2 mIU each of heparinase I, II, and III, and CS/DS chains were digested using 2 mIU chondroitinase ABC (Sigma) in 50 mM Tris/50 mM NaCl (pH 7.9). Resulting disaccharides were freeze-dried and 2-aminoacridone (AMAC)-labeled. HS and CS/DS disaccharides were separated by reverse-phase HPLC using a Zorbax Eclipse XDB-C18 column (4.6 × 100 mm, 3.5 μm; Agilent), equilibrated in 95% 0.1 M ammonium acetate/5% acetonitrile on an Agilent 1200 Series HPLC system. Disaccharides were eluted over 5–20% acetonitrile gradient at 0.2 ml/min. AMAC-labeled HS and CS/DS disaccharide standards (Iduron) were used for peak identification (see Appendix Fig S2 for peak assignment and identification of MPSII-specific peaks that do not correspond with standards and therefore were excluded from analysis). Total fluorescence was compared to known quantities of HS or CS/DS to calculate absolute amounts of each disaccharide. Correction factors were calculated as described (Appendix Table S1; Deakin & Lyon, 2008).

## Cytometric bead array

The levels of IL-1α, MCP-1, MIP-1α, and RANTES were measured in whole-brain and liver extracts using BD Cytometric bead array (CBA) Flex Set kits (BD Biosciences) according to the manufacturer's instructions, analyzed on a FACSCanto II flow cytometer, and processed using FCAP Array software (BD Biosciences).

## Real-time PCR experiments

Snap-frozen heart samples were defrosted in RNAlater-ICE (Thermo-Fisher) at $-20°C$, and RNA was extracted using TRIzol reagent (ThermoFisher). DNA was removed using TURBO DNA-free kit (ThermoFisher), and total RNA was reverse-transcribed with the High-Capacity cDNA Reverse Transcription kit (ThermoFisher). Quantitative polymerase chain reactions were performed with TaqMan Universal PCR Master Mix, using TaqMan gene expression assays for *Nppb* (*Mm01255770_g1*) *and Myh7* (*Mm00600555_m1*; ThermoFisher) on a StepOnePlus qPCR system (ThermoFisher). Glyceraldehyde-3-phosphate dehydrogenase (*Gapdh, Mm99999915_g1*) was used as a housekeeping gene. Data were analyzed using the $2^{-\Delta\Delta C_t}$ method.

## Immune response

ELISAs were performed on plasma samples serially diluted from 1:32 to 1:1,048,576 to measure human IDS-specific antibodies.

## The paper explained

### Problem
There is currently no effective cure for the severe neurodegeneration experienced by the majority of patients with mucopolysaccharidosis II.

### Results
Here, we present a hematopoietic stem cell gene therapy approach using a brain-targeted version of the missing enzyme IDS (IDS.ApoEII). We show complete normalization of working memory deficits, neuro-inflammation, and heparan sulfate storage, together with peripheral correction with IDS.ApoEII, providing a significant improvement over unmodified enzyme. The ApoEII peptide increases active IDS in plasma and increases enzyme uptake via different receptor types.

### Impact
This is an effective and scalable treatment for development into clinical trial for patients with MPS II. ApoEII peptide unexpectedly improves IDS effects through multiple mechanisms, not just by improving blood–brain barrier transport.

Plates were coated with 2 μg/ml Elaprase (Idursulfase; Shire, Basingstoke, UK) and detected with 5 μg/ml of goat anti-mouse IgG-HRP secondary antibody (Vector). A positive immune response to recombinant human IDS enzyme was produced by subcutaneously injecting naïve, age-matched MPS II mice with a mixture of Sigma Adjuvant System (Sigma) and 0.5 mg/kg Elaprase. Positive control plasma samples were obtained 14 days post-injection. Negative control plasma from uninjected MPS II mice was used as a baseline.

## Study approval

Animal experiments were ethically approved by the Manchester Research Ethics Committee and performed under UK Home Office regulations and PPL 40/3658.

## Statistics

Statistical analysis was performed using GraphPad Prism 7 software (La Jolla, CA, USA). Two-tailed parametric unpaired *t*-tests were applied for individual group comparisons. One-way or two-way ANOVAs were performed for multi-group analysis followed by Tukey's multi-comparisons test. Significance was set at $P < 0.05$. MPS II mutant mice and WT littermates were randomly assigned to control or transplant groups, although transplant group allocation was also partially determined by the number of donor animals and the amount of cells available for transplant. Behavioral analyses such as the rotarod could not be performed in an entirely blind fashion, because MPS II mice exhibit obvious skeletal and gait abnormalities. However, the Y-maze was recorded in a blind fashion and analyzed at a later time point from videography once all tests had been performed.

**Expanded View** for this article is available online.

## Acknowledgements

The authors gratefully acknowledge Professor Joseph Muenzer for providing the MPS II mouse model, Shire Plc for assisting with the transfer, and the staff

of the Manchester Biological Sciences Facility for their help and assistance. The Bioimaging Facility microscopes used in this study were purchased with grants from BBSRC, Wellcome, and the University of Manchester Strategic Fund. This work was completed with grants to BWB and HFEG from the Isaac Foundation and the National MPS Society (MPSII research grant 2015) and a grant to BWB and RJH from the Newlife Foundation (14-15/22). Special thanks go to Roger Meadows and Mike Broeders for their help with microscopy and figure design, respectively.

## Author contributions

HFEG designed the study, performed experiments, analyzed the data, and wrote the manuscript. AYL performed the immune response studies. JRC performed experiments. SFR, GMAF, and ZDS generated reagents and performed experiments. COL analyzed the data. RJH performed experiments, analyzed the data, and wrote the manuscript. BWB obtained the funds, designed the study, analyzed the data, and wrote the manuscript. All authors provided critical analysis of the manuscript.

## Conflict of interest

HFEG and BWB are co-inventors on a patent application (GB1701968.8) for the use of IDS.ApoEII for the treatment of MPS II.

## For more information

In addition, please find below a list of web links relevant to this article:

 OMIM: #309900

(i) https://mpssociety.org/

(ii) http://www.mpssociety.org.uk/

(iii) http://www.theisaacfoundation.com/

(iv) http://newlifecharity.co.uk/

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
