## [Review Process File · EMBO Molecular Medicine]

Brain targeted stem cell gene therapy corrects Mucopolysaccharidosis type II via multiple mechanisms

Hélène F. E. Gleitz, Ai Yin Liao, James R. Cook, Samuel F. Rowston, Gabriella M. A. Forte, Zelpha D'Souza, Claire O'Leary, Rebecca J. Holley, Brian W. Bigger

Review timeline:

Submission date:	29 November 2017
Editorial Decision:	22 December 2017
Revision received:	09 April 2018
Editorial Decision:	03 May 2018
Revision received:	10 May 2018
Accepted:	16 May 2018

Editor: Céline Carret

Transaction Report:

1st Editorial Decision

22 December 2017

Thank you for the submission of your manuscript to EMBO Molecular Medicine. We have now heard back from the two referees whom we asked to evaluate your manuscript. Although I was hoping to obtain a third evaluation, I am now proceeding based on the two consistent evaluations obtained so far as further delays cannot be justified.

Both Reviewers are mostly positive on your manuscript although they raise some important issues that require your action. I will not dwell into specifics, as their comments are very detailed and self-explanatory.

Therefore, we would welcome the submission of a revised version within three/four months for further consideration and would like to encourage you to address all the criticisms raised as suggested to improve conclusiveness and clarity. Please note that EMBO Molecular Medicine strongly supports a single round of revision and that, as acceptance or rejection of the manuscript will depend on another round of review, your responses should be as complete as possible.

I look forward to receiving your revised manuscript.

***** Reviewer's comments *****

Referee #1 (Comments on Novelty/Model System for Author):

see remarks to they author

Referee #1 (Remarks for Author):

Gleitz et al describe a lentiviral gene therapy approach for MPS II, a severe lysosomal storage disorder with multisystem pathology including the CNS. Especially the CNS symptoms (which result in early mental retardation) cannot be treated with current enzyme replacement therapy as recombinant IDS protein is not able to cross the blood brain barrier. There is therefore an unmet medical need to develop therapeutic strategies that can target the CNS. The current work describes a possible approach, in which hematopoietic stem cells are genetically corrected ex vivo using lentiviruses expressing epitope tagged IDS protein. The tag tested here, an ApoEII peptide, promotes the efficiency of the approach including passage of the blood brain barrier and stability of the enzymatically active form of IDS protein. A near-complete correction of lysosomal pathology in the CNS was obtained, next to complete recovery of peripheral symptoms. This is a solid piece of work consisting of extensive analyses of the various symptoms and their correction by the gene therapy approach. Functional cognitive analysis is guided by the previous work of the same authors. Although the effect of the MPS II mice on spontaneous alternation in the Y maze is very small, it is significant and can be restored by LV.IDS.ApoEII only. I have only minor comments, indicated below.

1. Figure 2F: can the authors speculate on why the enzyme activity in brain is similar between IDS and IDS.ApoEII, yet the latter clears GAGs better? Have the authors tried staining tissue sections for IDS protein to see whether there is differential targeting to different cell types/subcellular compartments?
2. Figure 3A,B: the analysis of HS derivatives has been performed using HPLC with co-elution with standards as identification method, referring to previous work from Holley et al (2011). In Holley et al., HS derivatives were analyzed in liver and brain tissue from mice with MPS I, which show a far larger increase in tissue HS compared to the MPS II mouse model. Peak identification was confirmed in that study using mass spectrometry. The question is whether the authors could provide some raw data on the analysis of the MPS II brain tissue, for examples showing some HPLC chromatograms as suppl info, and whether additional methods besides co-elation with standards were used to identify the peaks?
3. It is interesting to note that LV.IDS partially corrects HS accumulation in brain, and partially restores LAMP2 staining depending on the region. Considering the similar VCN of both vectors, this may suggest that a higher dosage of LV.IDS may yield similar effects compared to the current dosage of LV.IDS.ApoEII. Have the authors tested this, and/or could they elaborate on this?
4. It is also worth mentioning that some regions such as the amygdala do not show correction of lysosomal pathology (as monitored using LAMP2 staining) with LV.IDS.ApoEII
5. Is it possible to quantify the LAMP2 signal?
6. Figure 6L: please specify the negative control
7. Figure 7: last sentence of p11: I disagree with the notion that IDS and IDS.ApoEII have similar M6P status to that of HSCGT produced enzyme when using transfection into CHME3 cells. I understand the point of the authors regarding the cell type, but an in vivo lentiviral transduction cannot be compared to an in vitro transfection. For example, it is known that even slight changes in cell culture conditions can change the M6P status of recombinant proteins, let alone the potential effects of huge overexpression by transfection.
8. Fig 7I: the hypothesis of the authors that ApoEII may mediate binding via its HS site to LDLR and LRP1 receptors is interesting. However, untagged IDS seems to show a similar % of reduction in uptake following Heparinase or ApoEII treatment. How can the authors explain this?

9. In the uptake and transcytosis experiments, have the authors tested IDS enzyme activity as read out? Was IDS protein processed to its active form?

Referee #2 (Remarks for Author):

This manuscript describes the outcome of an hematopoietic stem cell gene therapeutic approach for MPS-II based on the use of a chimeric IDS enzyme containing a tandem repeat of the receptor-binding domain of the human apolipoprotein E (ApoE-II). The authors showed that IDS-ApoE-II provides a superior correction compared to IDS wild type. The authors conclude that such therapeutic superiority is due to the increased stability of the chimeric enzyme in the plasma and increased uptake and transcytosis across the blood brain barrier (BBB).

This is a comprehensive study analyzing several aspects of the therapeutic approach. However, the main issue of the study is why IDS-ApoE-II is therapeutically superior compared to the IDS WT. The authors claim that IDS-ApoE-II acts through different mechanisms including increased enzyme stability in the plasma and increased BBB crossing efficiency.

-Concerns regarding interpretation of data:

1) The ELISA experiment in the plasma (fig 7B) showed that IDS-ApoE-II is more active per unit of IDS protein. This means that the specific activity of IDS-ApoE-II is higher than IDS WT and not necessarily that IDS-ApoE-II is more stable (active site protected from degradation) compared to IDS WT (as the authors claim).

Enzyme stability should be measured by pulse & chase experiments

2) Enzyme clearance rate of IDS-ApoE-II and IDS WT is similar (fig 7A). Clearance rate is the outcome of different processes such as tissue uptake and enzyme stability. In principle the permanence of the enzyme in the plasma is negatively affected by the increased uptake from other organs and positively affected by increased stability. Therefore, the results showed in the figure 7A demonstrated that these two processes are acting in a way that leads to similar protein levels of IDS-ApoEII and IDS WT in the plasma.

-Concerns regarding conclusions

The enzyme activity in the brain of mice treated with LV-IDS-ApoEII and LV-IDS are very similar (3,7% and 3,4 % respectively). How such a small difference (0,3%) may explain the complete different effect on the phenotype (complete rescue upon IDS-ApoEII treatment vs mild improvement upon IDS treatment).

If IDS-ApoEII resulted in an increased enzyme uptake, BBB crossing efficiency and plasma stability then I would expect that the IDS activity measured in the total brain should be much more higher compared to IDS WT.

A more likely explanation could be that disease-relevant brain regions are targeted more efficiently when IDS-ApoEII is used. However this or other conclusions should be supported by data.

1st Revision - authors' response

09 April 2018

Referee #1:

1. Figure 2F: can the authors speculate on why the enzyme activity in brain is similar between IDS and IDS.ApoEII, yet the latter clears GAGs better? Have the authors tried staining tissue sections for IDS protein to see whether there is differential targeting to different cell types/subcellular compartments?

As suggested by the reviewer, we have now tried to perform immunohistochemistry on frozen brain sections in order to quantify staining and determine whether different cell types are specifically targeted. Unfortunately, there are few antibodies available against IDS and all those tested failed to

show any positive staining in all groups stained. The antibodies used were Novus biologicals NBP2-01745 at 1:100; Abcam AB85701 at 1:100; RnD systems AF2449, 1:50 (this is the antibody used in the ELISA assay as the capture antibody) used with and without antigen retrieval using citrate buffer at pH6. Therefore it is impossible to determine if there is any differential targeting at this stage. It is quite common for several of the available lysosomal enzyme antibodies to work only in ELISAs or Western blots, particularly MPS disease enzyme antibodies. Our hypothesis is that improved cell association, and uptake in LVIDS ApoEII treated mice improves the enzyme actually within cells and thus we get increased clearance. Enzyme turnover within cells could account for the similarities in activity. We have added a comment to this effect into discussion

2. Figure 3A,B: the analysis of HS derivatives has been performed using HPLC with co-elution with standards as identification method, referring to previous work from Holley et al (2011). In Holley et al., HS derivatives were analyzed in liver and brain tissue from mice with MPS I, which show a far larger increase in tissue HS compared to the MPS II mouse model. Peak identification was confirmed in that study using mass spectrometry. The question is whether the authors could provide some raw data on the analysis of the MPS II brain tissue, for examples showing some HPLC chromatograms as suppl info, and whether additional methods besides co-elution with standards were used to identify the peaks?

HPLC chromatograms of HS from WT and MPSII alongside disaccharide standards used for peak identification have now been added to the manuscript as supplementary figure 2. 6 peaks were common to WT and MPSII samples and co-elute with the known standards, similar to what we have published previously (Holley et al., J Biol Chem. 2011; Holley et al., Brain 2017). Unidentified peaks are present in MPSII, however these provide a minor contribution to the overall levels of HS and have not been included in analyses. Other groups have worked to characterise these peaks (Lawrence et al., Nat Chem Biol. 2012) and these are likely to represent non-reducing end disaccharides.

3. It is interesting to note that LV.IDS partially corrects HS accumulation in brain, and partially restores LAMP2 staining depending on the region. Considering the similar VCN of both vectors, this may suggest that a higher dosage of LV.IDS may yield similar effects compared to the current dosage of LV.IDS.ApoEII. Have the authors tested this, and/or could they elaborate on this?

We used an MOI of 100 for either LV.IDS or LV.IDS.ApoEII vectors to transduce MPS II haematopoietic stem cells for 24 hours prior to transplant. This MOI results in almost equivalent VCNs in bone marrow of ~4 copies per cell at the time of transplant, which is clinically relevant. Slightly lower VCNs were typically recorded 6 months post-transplant in LV.IDS-ApoEII compared to LV.IDS in haematopoietic organs. Haematopoietic repopulation of the bone marrow is driven by clonal expansion of several transduced and untransduced stem cell clones. Assuming no toxicity of the vector or transgene (that we do not observe here), this process is driven by chance – thus the relative VCN of the BM and blood can change as these clones expand and contract, as is seen here. IDS.ApoEII has reduced VCN per cell in the BM than IDS, and this is reflected by slightly lower enzyme levels in these organs. In the brain, we are relying on monocytic cells crossing the BBB and engrafting as macrophages in the brain itself. Hence the VCN is much lower, but is similar for both vectors, as is the enzyme level, as the reviewer points out. However, the additional HS clearance, inflammation clearance etc in the brain by IDS.ApoEII suggests a more potent vector. We could certainly achieve a higher dosage of LV.IDS, using a double transduction of HSCs with LV.IDS, and this may result in similar effects to LV.IDS.ApoEII in the brain. However, this will most likely require a significant increase in haematopoietic VCN for LV.IDS to achieve this. Clinically, it is desirable to maintain as few integrations into HSCs as possible to achieve correction, whilst minimising potential side effects of integrated lentiviral vector. Thus being able to correct the brain by achieving a BM VCN of 2 with LV.IDS.ApoEII is much more desirable than having to achieve a VCN of >4 (and potentially double this) with LV.IDS to achieve the same brain effect. We have added in lines to the discussion to cover some of this aspect.

4. It is also worth mentioning that some regions such as the amygdala do not show correction of lysosomal pathology (as monitored using LAMP2 staining) with LV.IDS.ApoEII

We wish to clarify to the reviewer that we observe correction of lysosomal pathology in the amygdala in the LV.IDS.ApoEII group, as shown in figure 4a and figure 5a. Note that these are

representative images, thus there is some positive staining in 4a in the amygdala. However, quantification of the LAMP2 signal in the amygdala has been added as suggested by this reviewer's next comment in figure 4 and figure 5, which represents the entire group of IHC stained mice. This now clearly shows that LAMP2 staining is reduced to WT levels with LV.IDS.ApoEII. Note that as all cells have lysosomes – there is always some LAMP2 staining even in WT cells, just at a much lower level to MPSII cells. We have added quantification to fig 4 and 5 and clarified the section in results.

5. Is it possible to quantify the LAMP2 signal?

The LAMP2 signal has been quantified in the cortex and amygdala. Figure 3C has now been moved to figure 4 and quantification added to this figure. All following figures have been re-numbered to account for this change.

6. Figure 6L: please specify the negative control

The negative control used was plasma from an uninjected MPSII mouse. This detail has now been added to the 'Materials and methods' section and to the figure legend.

7. Figure 7: last sentence of p11: I disagree with the notion that IDS and IDS.ApoEII have similar M6P status to that of HSCGT produced enzyme when using transfection into CHME3 cells. I understand the point of the authors regarding the cell type, but an *in vivo* lentiviral transduction cannot be compared to an *in vitro* transfection. For example, it is known that even slight changes in cell culture conditions can change the M6P status of recombinant proteins, let alone the potential effects of huge overexpression by transfection.

We thank the reviewer for pointing out this discrepancy. We accept that eluding that M6P modification of the IDS proteins produced *in vitro* versus *in vivo* are equivalent may not be correct. Therefore we have removed the comment about the M6P status of the protein from the description of Figure 8 (originally figure 7). In designing this experiment, we have tried to make this scenario as close as possible to the *in vivo* situation, but we understand the reviewer's concerns.

8. Fig 7I: the hypothesis of the authors that ApoEII may mediate binding via its HS site to LDLR and LRP1 receptors is interesting. However, untagged IDS seems to show a similar % of reduction in uptake following Heparinase or ApoEII treatment. How can the authors explain this?

We agree with the reviewer that there appears to be a very small decrease with either heparinase or ApoE in the IDS group, but this is not significantly different from enzyme only cells, whilst the decrease mediated by heparinase or ApoE in the IDS.ApoEII group is larger and also significant compared to enzyme only control. We have clarified this in the results.

9. In the uptake and transcytosis experiments, have the authors tested IDS enzyme activity as read out? Was IDS protein processed to its active form?

The IDS protein used for transcytosis experiments was obtained from media of CHME3 transfected with IDS/IDS.ApoEII cDNA. An IDS activity assay was performed to ensure that IDS protein was processed to its active form prior to addition to bEND.3 cells. Quantification by ELISA was then used to enable equal quantities of IDS or IDS-ApoEII protein to be added to the bEND.3 cells in transcytosis assays. In transcytosis and uptake assays, the amounts of IDS enzyme activity detected afterwards are too small to obtain accurate values above background, hence we opted to perform an IDS ELISA, which has greater sensitivity.

Referee #2:

-Concerns regarding interpretation of data:

1) The ELISA experiment in the plasma (fig 7B) showed that IDS-ApoE-II is more active per unit of IDS protein. This means that the specific activity of IDS-ApoE-II is higher than IDS WT and not

necessarily that IDS-ApoE-II is more stable (active site protected from degradation) compared to IDS WT (as the authors claim). Enzyme stability should be measured by pulse & chase experiments.

We understand the referee's concerns and have therefore measured intracellular enzyme activity (by enzyme activity assay) and IDS protein from the same sample (by ELISA), which is now included in the Supplementary data. Additionally, we do not have a purified enzyme product, as the enzyme is secreted into media, and therefore pulse/chase experiments would not be feasible at this point and are likely beyond the scope of this manuscript.

However, we performed additional experiments to specifically investigate levels of enzyme activity per unit of protein in CHME3 cells and whether IDS.apoEII enzyme was more active per unit of IDS protein unit (supplementary figure 5). IDS and IDS.ApoEII enzyme were produced by transfection in CHME3 human microglial cells into the culture media. Secreted and intracellular enzyme levels were standardised by ELISA and subsequently analysed for IDS enzyme activity (n=4 per group). We did not observe any significant differences between IDS and IDS.ApoEII in enzyme activity per unit of protein, either intracellularly or extracellularly. However, this experiment was performed *in vitro* and it may be significantly different to the *in vivo* effect, which remains difficult to investigate. Thus whilst we cannot discount the possibility that the specific activity of the IDS.ApoEII enzyme is increased *in vivo*, it seems more likely that it has improved plasma stability from the above experiment. We have adjusted the text to reflect both of these possibilities and reduced our claim to noting that there is more active enzyme in plasma in abstract, results and discussion and that it could be due to either effect.

2) Enzyme clearance rate of IDS-ApoE-II and IDS WT is similar (fig 7A). Clearance rate is the outcome of different processes such as tissue uptake and enzyme stability. In principle the permanence of the enzyme in the plasma is negatively affected by the increased uptake from other organs and positively affected by increased stability. Therefore, the results showed in the figure 7A demonstrated that these two processes are acting in a way that leads to similar protein levels of IDS-ApoEII and IDS WT in the plasma.

We thank the reviewer for their insight and agree that this effect could be mediated by a combination of increased uptake/degradation, depending on the enzyme. We would like to further highlight that the clearance experiment is performed using a bolus injection of unpurified enzyme injected into the bloodstream and it is unclear whether this reflects the situation of chronic low level steady state enzyme delivery, which is obtained with HSCGT. It also remains unknown whether enzyme secretion is different in blood cells such as monocytes, compared to microglial cells. We have added wording to the manuscript to reflect this uncertainty and have made our main mechanistic claim based on the improved cell association and uptake via multiple receptors with ApoEII that we observe in bEND3 cells.

-Concerns regarding conclusions

The enzyme activity in the brain of mice treated with LV-IDS-ApoEII and LV-IDS are very similar (3,7% and 3,4 % respectively). How such a small difference (0,3%) may explain the complete different effect on the phenotype (complete rescue upon IDS-ApoEII treatment vs mild improvement upon IDS treatment).

If IDS-ApoEII resulted in an increased enzyme uptake, BBB crossing efficiency and plasma stability then I would expect that the IDS activity measured in the total brain should be much more higher compared to IDS WT.

A more likely explanation could be that disease-relevant brain regions are targeted more efficiently when IDS-ApoEII is used. However this or other conclusions should be supported by data.

We thank the reviewer for their thoughtful comments. We agree that it is surprising that such small changes could make such drastic changes to the neuropathology, yet we have extensive evidence that the addition of the ApoEII peptide is critical to this process.

Firstly, we did not observe obvious regionality differences in storage clearance between the vectors in the brains of treated mice. LV.IDS cleared less LAMP2 from all brain regions than LV.IDS.APoEII. We have seen a widespread decrease of LAMP2 in all different disease-specific areas that we have examined, hence it is more likely that IDS.ApoEII enzyme has a global effect on the brain rather than on specific areas. Notably, there are minor regional differences in clearance in

different brain regions (this is true of both vectors), but LV.IDS.ApoEII mediates the most complete global clearance of LAMP2 in the brain that we have observed in any lysosomal disease model that we have studied.

To clarify this, the LAMP2 signal has been quantified in the cortex and amygdala. Figure 3C has now been moved to figure 4 and LAMP2 quantification added to figure 4 and figure 5. All following figures have been re-numbered to account for this change.

Secondly, most of the cells that produce enzyme in the brain are monocyte-derived macrophages from bone marrow. We observed similar VCN in the brain between IDS and IDS.ApoEII groups, indicating that similar numbers of cells are present and produce similar levels of enzyme. The main difference is that IDS.ApoEII enzyme is taken up into bEND3 cells more efficiently (Fig 7), and by extension may also be taken up into brain cells more efficiently, thereby increasing the therapeutic effect. Importantly, enzyme activity measured in the brain is performed on whole brain homogenate, and it is therefore difficult to determine where the enzyme is precisely contained and active. Hence, although similar levels of enzyme are detected in both IDS and IDS.apoEII groups in the brain, enzyme localisation within brain cells and brain parenchyma could be significantly different. Unfortunately efforts to stain IDS enzyme have not been successful – see first comment to reviewer 1. Our hypothesis is that improved cell association, and uptake in LV.IDS.ApoEII treated mice improves the enzyme actually within cells and thus we get increased clearance. Enzyme turnover within cells could account for the similarities in activity. We have added a comment to this effect in the discussion.

[Deleted additional text as referred to unpublished material]

2nd Editorial Decision

03 May 2018

Thank you for the submission of your revised manuscript to EMBO Molecular Medicine. We have now received the enclosed reports from the referees that were asked to re-assess it. As you will see the reviewers are now supportive and I am pleased to inform you that we will be able to accept your manuscript pending editorial final editorial amendments.

Please submit your revised manuscript within two weeks. I look forward to seeing a revised form of your manuscript as soon as possible.

***** Reviewer's comments *****

Referee #1 (Remarks for Author):

The authors have satisfactorily addressed my comments.

Referee #2 (Remarks for Author):

The authors addressed sufficiently all the points raised in the first round of review.

2nd Revision - authors' response

10 May 2018

Authors made the requested editorial changes.

Corresponding Author Name: Brian Bigger
Journal Submitted to: EMBO Molecular Medicine
Manuscript Number: EMM-2017-08730-V2